# GROUP PREFERENCE OPTIMIZATION: FEW-SHOT ALIGNMENT OF LARGE LANGUAGE MODELS

**Siyan Zhao, John Dang, Aditya Grover**
Department of Computer Science, University of California, Los Angeles
{siyanz,john.dang,adityag}@cs.ucla.edu

## ABSTRACT

Many applications of large language models (LLMs), ranging from chatbots to creative writing, require nuanced subjective judgments that can differ significantly across different groups. Existing alignment algorithms can be expensive to align for each group, requiring prohibitive amounts of group-specific preference data and computation for real-world use cases. We introduce Group Preference Optimization (GPO), an alignment framework that steers language models to preferences of individual groups in a few-shot manner. In GPO, we augment the base LLM with an independent transformer module trained to predict the preferences of a group for the LLM generations. For few-shot learning, we parameterize this module as an in-context autoregressive transformer and train it via meta-learning on several groups. We empirically validate the efficacy of GPO through rigorous evaluations using LLMs with varied sizes on three human opinion adaptation tasks. These tasks involve adapting to the preferences of US demographic groups, global countries, and individual users. Our results demonstrate that GPO not only aligns models more accurately but also requires fewer group-specific preferences, and less training and inference computing resources, outperforming existing strategies such as in-context steering and fine-tuning methods. [1]

*Warning: This paper contains qualitative examples that may be viewed as offensive or harmful.*

## 1 INTRODUCTION

Large Language Models (LLMs) are increasingly being employed for a wide variety of domains, with use-cases including creative writing, chatbots, and semantic search among others (Touvron et al., 2023b; Taori et al., 2023; Ouyang et al., 2022; Bai et al., 2022a;b; Brown et al., 2020). Many of these applications are inherently subjective and require generations that cater to different demographics, cultural and societal norms, or simply individual preferences (Hartvigsen et al., 2022; Zhang et al., 2023; Solaiman & Dennison, 2021; Blodgett et al., 2020; Dunbar et al., 1997). By virtue of their large-scale training, current language models are exposed to diverse data that allows them to *represent* a multitude of such opinions (Glaese et al., 2022; Durmus et al., 2023; Santurkar et al., 2023). However, expressing these diverse opinions requires steering the LLM generations to user requirements. This brings forth the key question studied in this work:

*How do we efficiently adapt LLMs to align closely with the opinions of specific interest groups?*

Broadly, prior work has explored two modes of steering language models, which trade-off training complexity with test-time engineering. On one end, prompt engineering approaches avoid explicit modifications to the parameters of the language model and elicit desired behavior by crafting a suitable prompt. Often, the prompt is augmented with a few in-context examples (Brown et al., 2020; Taori et al., 2023; Chowdhery et al., 2022). While prompting approaches are attractive as they have no additional training complexity over the base model, prompt engineering can be quite tedious and empirically poor when the desired behaviors are more complex (Zhou et al., 2022; Reynolds & McDonell, 2021; Qin & Eisner, 2021; Lester et al., 2021). For example, Santurkar et al. (2023) show

---

[1] Our code is available at the project website: https://siyan-zhao.github.io/llm-gpo/

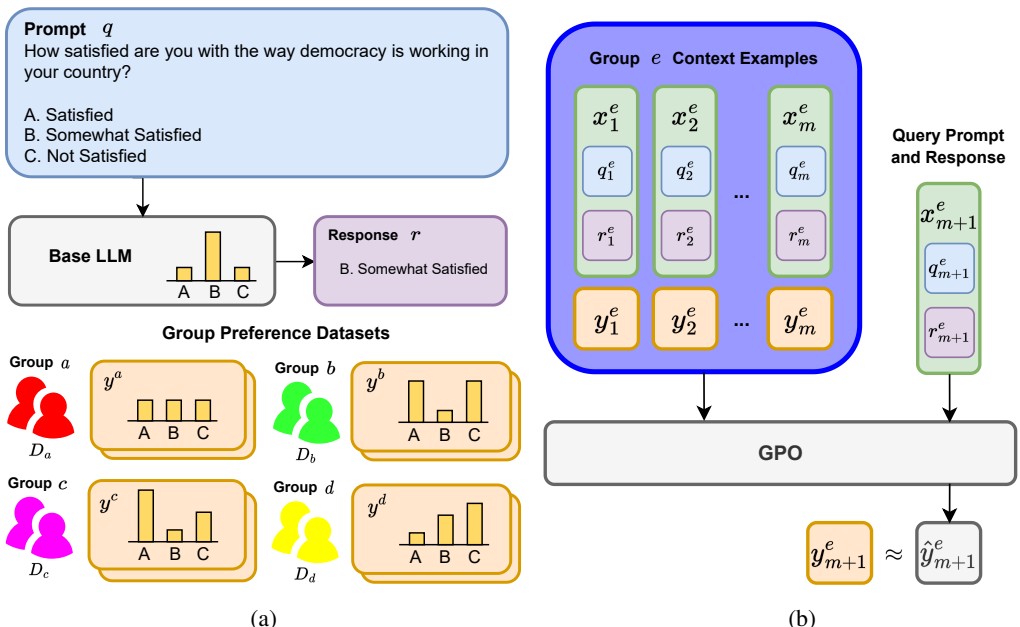

Figure 1: Overview of GPO. **Left:** Group alignment aims to steer pretrained LLMs to preferences catering to a wide range of groups. For each group $g$, we represent its preference dataset as $\mathcal{D}_g = \{(x_1^g, y_1^g), \ldots, (x_n^g, y_n^g)\}$. Here, $y_i^g$ signifies the preference of group $g$ for a pair of given prompt $q_i^g$ and response $r_i^g$, while $x_i^g$ is its LLM representation obtained with $\pi_{\text{emb}}(q_i^g, r_i^g)$. **Right:** Once trained, GPO provides a few-shot framework for aligning any base LLM to a test group given a small amount of in-context preference data.

that LLMs over-emphasize opinions from privileged demographics and are challenging to rectify via in-context prompting approaches.

On the other end, various kinds of alignment approaches have been proposed that seek to augment or finetune the language model with an additional reward or scoring model. These approaches can steer the model to achieve complex behaviors such as honesty, helpfulness, and harmlessness (Ouyang et al., 2022; Bai et al., 2022a; Glaese et al., 2022; Bansal et al., 2023; Askell et al., 2021; Song et al., 2023; Bai et al., 2022b; Thoppilan et al., 2022; Wang et al., 2022), but come at the cost of additional complexity in gathering sufficient supervision to train reward models and subsequent finetuning. As a result, existing alignment approaches, such as PPO (Schulman et al., 2017), DPO (Rafailov et al., 2023), and Best-Of-N, are not designed to efficiently align LLMs when the number of target groups is large and supervision for each group is limited.

We introduce *Group Preference Optimization* (GPO), a few-shot framework for aligning Large Language Models to opinions and preferences of desired interest group(s). The key idea in GPO is to view the alignment of an LLM policy as a few-shot adaptation problem within the embedded space of an LLM. Specifically, GPO augments an arbitrary base LLM with an independent few-shot preference module. This module is parameterized via an independent transformer and trained to explicitly perform in-context supervised learning to predict preferences (targets) given joint embeddings (inputs) of prompts and corresponding LLM responses. The use of embeddings guarantees that the preference module can effectively process in-context examples where each example is itself a potentially long sequence of prompt and generated response. In-context learning further provides the ability to efficiently adapt to new, unseen groups at test-time with only a handful of examples. See Figure 1 for an illustration. Finally, we incorporate various architectural design choices to guarantee permutation-specific inductive biases, building on recent work in in-context learning over datasets (Nguyen & Grover, 2022). Once learned, the learned module can serve as a drop-in replacement for a reward or preference function for policy optimization and re-ranking algorithms.

In our experiments, we validate the effectiveness of GPO for aligning language models to the opinions of 22 diverse US demographic groups in the OpinionQA dataset (Santurkar et al., 2023) and

14 global countries in the GlobalOpinionQA dataset (Durmus et al., 2023). We consider 2 base language models of different sizes: Alpaca 7B (Taori et al., 2023), an instruction-tuned version of the LLaMA (Touvron et al., 2023a) 7B model, and the recent Llama2 13B chat (Touvron et al., 2023b), which has been fine-tuned on a large dataset of human preferences for helpfulness and safety. Empirically, we test GPO against a variety of prompting and finetuning baselines. On average, GPO surpasses the top-performing baselines by 7.1% when adapting to 22 US demographic groups in OpinionQA, and by 8.4% when aligning with 14 global countries in GlobalOpinionQA. Furthermore, GPO performs most effectively in adapting to individual preferences compared to other baselines.

## 2 GROUP PREFERENCE OPTIMIZATION

### 2.1 PROBLEM SETUP

A large language model (LLM) expresses a probability distribution over natural language, denoted as $\pi$. To accomplish any task, such as question answering or summarization, a user crafts a suitable query $q$ and prompts the LLM to generate a response $r$ obtained via sampling from the conditional distribution $\pi(\cdot \mid q)$. Rather than decoding responses from a single distribution $\pi(\cdot \mid q)$, our goal in this work is to align the language model to the preferences of a desired target group $g^* \in G$. Here, we adopt a fairly general definition of a *group* to refer to any collection of agents (e.g., demographic groups, individual personas), and we use $G$ to denote the space of all possible groups. For training, we assume that we are given access to preference datasets for a finite set of training groups $G_{\text{train}}$. In practical applications, the number of groups can be large (e.g., different demographics and cultures) while the amount of preference data for each group is generally small.

### 2.2 RELATED WORK

Existing approaches for steering LLMs are challenging to apply for group alignment, especially when the underlying groups are complex and per-group supervision is scarce. Below, we summarize key approaches and their trade-offs, which will also serve as baselines in our experiments. We provide additional discussion of related work in Appendix I.

**Prompt Engineering:** These approaches modify the input prompt $q \rightarrow q'$ to guide the LLM towards a group-aligned distribution (Jiang et al., 2023; Hwang et al., 2023; Deshpande et al., 2023). Techniques include meta-data utilization, where group-specific meta-data, are appended to the input prompt. Further, the engineered prompts can be improved via in-context few-shot prompting, in which the prompt is concatenated with examples of desired behavior. Given the flexibility of language, even a preference dataset $D_g$ could be converted into in-context examples for improving the prompt. Prompt engineering approaches are computationally efficient as they involve no training, but designing the prompt itself can be a tedious task that relies on heuristics (Zhou et al., 2022; Lester et al., 2021; Qin & Eisner, 2021), which are not guaranteed to transfer well across different LLMs. Finally, it has been shown prompt engineering has limited gains in aligning LLMs to complex groups on challenging survey datasets (Santurkar et al., 2023; Durmus et al., 2023).

**Gradient-based Alignment:** Algorithms that fine-tune the base Large Language Model (LLM) or augment it with additional models have successfully aligned LLMs to complex behaviors like honesty, harmfulness, and helpfulness. Broadly, there are two main classes of methods. The first involves supervised learning, using a dataset of responses from a target group for fine-tuning, as demonstrated in (Ouyang et al., 2022; Ziegler et al., 2019). This method is straightforward but often suffers from limited generalization and requires extensive group-specific data. The second class uses explicit human-derived preference data to train reward models for response filtering, re-ranking (e.g., Best-of-N, importance weighting (Grover et al., 2019)), or reinforcement learning optimization (e.g., PPO (Ouyang et al., 2022; Schulman et al., 2017)), which may pose challenges in hyperparameter tuning and stability (Sun et al., 2023; Santacroce et al., 2023). Newer methods focus on direct optimization of preferences to enhance stability in RL techniques (Rafailov et al., 2023; Song et al., 2023). These approaches typically require access to large preference datasets. Our method, GPO, is designed to explicitly align with various interest groups under limited supervision constraints, positioning it within the explicit alignment framework.

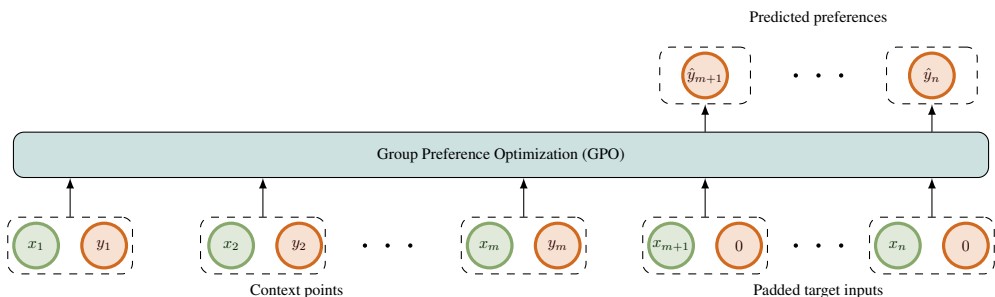

Figure 2: Illustration of the GPO architecture for a sequence of $n$ points, with $m$ context points and $n - m$ target points. The context $(x_{1:m}, y_{1:m})$ serves as few-shot conditioning for GPO. GPO processes the full sequence using a transformer and predicts the preference scores $\hat{y}_{m+1:n}$.

## 2.3 PROPOSED METHOD

We desire an alignment approach that generalizes to a wide variety of groups, even when constrained by the amount of per-group supervision. Accordingly, we view group alignment as a few-shot learning problem and cast it in the framework of in-context meta-learning. For each training group $g \in G_{\text{train}}$, we represent its preference dataset as $\mathcal{D}_g = \{(x_1^g, y_1^g), \ldots, (x_n^g, y_n^g)\}$ where $y_i^g$ denotes the preference of group $g$ to a pair of input prompt query $q_i^g$ and LLM response $r_i^g$, and $x_i^g$ denotes the LLM representation of the concatenation of the prompt query and LLM response $x_i^g = \pi_{\text{emb}}(q_i^g, r_i^g)$. Here, $\pi_{\text{emb}}$ can be the language model embedding function or an identity function that maintains the input's raw textual format. Note that while the inputs $x^g$ can be shared across different groups (e.g., universal surveys), the preferences are different for each group. At test-time, our goal will be to steer the default LLM distribution to a new distribution, say $\pi_{g^*}$, given a preference dataset $\mathcal{D}_{g^*}$ for the target query group $g^*$. For brevity of presentation, we consider the preference to be a real-valued scalars. Our framework extends to other kinds of responses and preferences, such as short-answer questions (e.g., MCQs) and relative pairwise responses, as discussed in Appendix H.

Given the above setup, we design GPO to perform group alignment by learning a few-shot preference model that augments the base LLM, as shown in Algorithm 1. Once learned, we can use it to update the LLM via any standard preference optimization or reweighting algorithm (e.g., PPO, Best-of-N). Specifically, we parameterize GPO via a transformer and train it to perform in-context learning on the training preference datasets. Given a training group $g \in G_{\text{train}}$, we randomly split its preference dataset $\mathcal{D}_g$ into a set of $m$ context points and $n - m$ target points, where $n = |\mathcal{D}_g|$ is the size of the preference dataset for group $g$. Thereafter, GPO is trained to predict the target preferences $y_{m+1:n}^g$ given the context points $(x_{1:m}^g, y_{1:m}^g)$ and target inputs $x_{m+1:n}^g$. Mathematically, we can express the objective as:

$$L(\theta) = \mathbb{E}_{g,m} \left[ \log p_\theta(y_{m+1:n}^g \mid x_{1:n}^g, y_{1:m}^g) \right] \tag{1}$$

where the training group $g \sim G_{\text{train}}$ and context size $m$ are sampled uniformly. $\theta$ represents the parameters of our model. Figure 2 shows an illustration. For decoding, we make the conditional independence assumption, where we assume that the target preferences are independent of each other given the context samples and the target inputs:

$$L(\theta) = \mathbb{E}_{g,m} \left[ \sum_{i=m+1}^{n} \log p_\theta(y_i^g \mid x_{1:n}^g, y_{1:m}^g) \right] \tag{2}$$

In our preliminary experiments, we also investigated alternatives which model the dependencies. We did not find any noticeable improvements and hence use Eq. 2 for the rest of the paper.

Following Nguyen & Grover (2022), we can modify the transformer architecture in GPO to explicitly account for permutation invariance conditioning over in-context examples. In particular, we discard the positional encodings commonly found in standard transformer architectures. However, this loses the pairwise relations between $(x_i, y_i)$. To solve this, we concatenate each pair $(x_i, y_i)$ into a single token to inform the transformer of their pairwise relation. For the target inputs, we pad the $x_i$'s with a dummy token (e.g., 0). Finally, we employ a masking strategy where the context pairs can self-attend to each other, whereas the padded targets can only attend to the context points and

not to other target points to follow the conditional independence assumption in Eq. 2. GPO satisfies the properties of context invariance (Property 1.) and target equivalence (Property 2.).

Note that even though GPO uses in-context learning, it is distinct from in-context prompting a base LLM. The latter does not update the parameters of the base LLM and requires examples of desired text generations. On the other hand, GPO learns a few-shot model which augments the base LLM and only requires preferences of users for the LLM generations. That said, both these schemes are complementary to each other as we can use any engineered prompt (e.g., with in-context examples) as a drop-in replacement for the default prompt used in the inputs $x$.

**Scaling to long dataset contexts.** One challenge with GPO is that the effective sequence length for the transformer can grow significantly if we use raw representations of prompts and responses within each input $x$. This can degrade performance and efficiency significantly. To overcome this challenge, we propose to use embedded representations of text within $x$, as LLM representations can contain sufficient information for solving tasks (Bhatia et al., 2023). In particular, we first concatenate the prompt and response and compute their joint embedding $\pi_{\text{emb}}(q_i^g, r_i^g)$ using the base LLM. We explored different techniques for extracting the joint embeddings from the base LLM, as detailed in the ablation study in Appendix D, and found it best to use the average embedding of all the tokens in the input.

---

**Algorithm 1** *Group Preference Optimization* (GPO)

---

1: **Input:** LLM embeddding function $\pi_{\text{emb}}$; Preference datasets $\mathcal{D}_g \ \forall g \in G_{\text{train}}$.
2: Initialize GPO transformer with parameters $\theta$.
3: For all $g \in G_{\text{train}}$, cache embedded pairs $(x_i^g, y_i^g)$ in $\mathcal{D}_g^{\text{emb}}$ where $x_i^g = \pi_{\text{emb}}(q_i^g, r_i^g)$.
4: **repeat**
5:      Sample training group $g \in G_{\text{train}}$.
6:      Sample context size $m \sim \text{Uniform}[1, n-1]$ where $n = |D_g|$.
7:      Split $\mathcal{D}_g^{\text{emb}}$ randomly into $m$ context $(x_{1:m}^g, y_{1:m}^g)$ and $(n-m)$ target $(x_{m+1:n}^g, y_{m+1:n}^g)$ pairs.
8:      Predict target preferences $y_{m+1:n}^g$ using context $(x_{1:m}^g, y_{1:m}^g)$ and padded targets $(x_{m+1:n}^g, 0)$.
9:      Update $\theta$ to minimize in-context loss function $L(\theta)$ in Eq. 2.
10: **until** convergence
11: **Output:** GPO transformer with learned parameters $\theta$

---

## 3 EXPERIMENTS

**Datasets.** While GPO is general-purpose and can be applied broadly to many language model use cases, our work is focused on benchmarks which reflect a diverse landscape of human preferences. Quantitatively evaluating the diverse opinions through open-ended questions (e.g., creative writing) is inherently complex, and often demands expensive human labels. In contrast, closed-ended responses (e.g., multiple-choice questions) offer a standardized means of capturing diverse opinions, thus reducing ambiguity and noise in evaluation. Survey datasets have been used in prior work (Santurkar et al., 2023; Durmus et al., 2023) to demonstrate the weaknesses of current LLMs in catering to diverse populations, and hence can be effectively used to benchmark progress in group alignment.

We benchmark group alignment on 2 recent survey datasets: (1) *OpinionQA* (Santurkar et al., 2023), which spans 22 US demographic groups (e.g. income, political ideology, race, and sex) across 500 multiple-choice questions and (2) *GlobalOpinionQA* (Durmus et al., 2023), which contains multiple-choice questions answered by participants from 14 countries, amounting to 2,554 questions which cover various topics including politics, media, technology, religion, race, and ethnicity. Survey questions are shared across different groups, so we use $x_i$ (and not $x_i^g$) for brevity henceforth. Detailed dataset descriptions can be found in Appendix B.

Next, we construct group $g$ preference dataset $\mathcal{D}_g$ from the survey data. Let $Q$ be the set of all survey questions and $G$ be the groups participating in the survey. Consider a survey question $q \in Q$, with $T$ unique answer options. Each option can be interpreted as a response $r$, yielding a set of $T$ viewpoints $\{x_i\}_{i=1}^T = \{\pi_{\text{emb}}(q, r_i)\}_{i=1}^T$. The preference score $y_i^g$ for the viewpoint $x_i$ is obtained by

aggregating the survey responses given to $(q, r_i)$ from group $g$. These scores are normalized within each question to form the group preference distribution vector $P_g(q) = [y_1^g, ..., y_T^g]$ for question $q$, such that $\sum_{i=1}^{T} y_i^g = 1$. Repeating this process for all $n$ questions in $Q$ yields $\mathcal{D}_g$. During training and testing, all viewpoints from the same question belong to either the context or target set. Finally, we apply a softmax layer to predictions for each question, yielding normalized preference scores for each survey question in the target set.

**Evaluation Metric.** To rigorously assess the degree of alignment between two opinion distributions, $P_1$ and $P_2$, we calculate the *Alignment Score*, denoted as $\mathcal{A}(P_1, P_2; Q)$ over a set of questions $Q$. This metric employs a similarity function *Sim*:

$$\mathcal{A}(P_1, P_2; Q) = \frac{1}{|Q|} \sum_{q \in Q} Sim(P_1(q), P_2(q)) \tag{3}$$

For the OpinionQA dataset (Santurkar et al., 2023) with its ordinal answers, we employ the one-dimensional Wasserstein Distance as our similarity metric. Conversely, for the GlobalOpinionQA dataset, which often presents non-ordinal answer structures, we use the Jensen-Shannon Distance as suggested by the original paper. Further details are available in Appendix B.

**Base Large Language Models.** We use two different-sized LMs as our base models for baselines and GPO. The first, Alpaca-7B Taori et al. (2023), is an instruction-tuned variant of the Llama-7B (Touvron et al., 2023a), crafted using 52K instruction-response pairs. The second, Llama2-13B chat version, is finetuned over 1M human preferences for helpfulness and safety. For baseline methods requiring updates of model weights, we use low-rank adaptation (LoRA) (Hu et al., 2021).

**Baselines.** We compare our method against extensive baseline approaches as introduced below. For a detailed description of the baselines, refer to the Appendix F. (1) **Uniform Distribution** assumes equal preference scores for all options; (2) *LM Base* following Santurkar et al. (2023); Durmus et al. (2023), gets the LM's default opinion distribution $P_\pi(q)$ by extracting and normalizing prediction scores for answer choices; (3) *LM Steered* uses prompting strategies to convey group information to the LM (examples in Appendix K); (4) *Few-shot Prompt* appends a few examples showing a group's preferences for $m$ context questions to the prompt, where $m$ is constrained by the LM's context window size and $c_g$ includes the context samples $\{x_i, y_i\}_{i=1}^{m}$ (see Figure 8 in the Appendix); (5) *SFT per group* fine-tunes the LM separately for each group $g$ with a maximum likelihood loss on augmented training examples created by sampling responses $r$ according to the preference distribution $P_g(q)$; (6) *Reward Model* trains a per-group reward model by adding a linear MLP head on a base LLM and training it on $m$ context samples $\{x_i, y_i^g\}_{i=1}^{m}$ with MSE loss to predict preference scores; and (7) *In-Context Finetune* investigates few-shot in-context alignment ability by partitioning the group set into meta-train/test sets, splitting training questions into context/query, supplementing each query $q$ with a context $c_g$ of $m$ ground truth preferences, and fine-tuning the LM with maximum likelihood loss where responses are sampled according to $P_g(q)$.

## 3.1 RESULTS AND DISCUSSION

**Adapting to US demographics in OpinionQA.** We conducted experiments with three distinct meta-train and meta-test splits, allocating 40%, 60%, and 80% of the 22 US demographics groups respectively as the meta-train groups $G_{train}$. This group split was consistent with the *In-context Finetune* baseline and GPO. For other baselines that operate on a per-group basis, we calculated the alignment score for the meta-test groups and present results averaged over three random seeds.

Our results are presented in Figure 3. Alpaca-7b base model exhibit alignment scores that are similar to the alignment score of a uniform distribution. This does not necessarily imply an absence of biases, as averaging across groups could obscure biases towards certain demographics. Prior work has found LMs may disproportionately over-represent some groups and under-represent others (Santurkar et al., 2023). However, Llama2-13b-chat base model exhibits a lower alignment score as compared to the uniform distribution. This might be attributed to its fine-tuning for safety, causing the model to lean towards the least harmful option, which can be seen from the qualitative examples in Appendix J. When we incorporate group information into the LMs, we deploy various prompting strategies—QA, BIO, and PORTRAY—to convey this information (see Appendix K for examples).

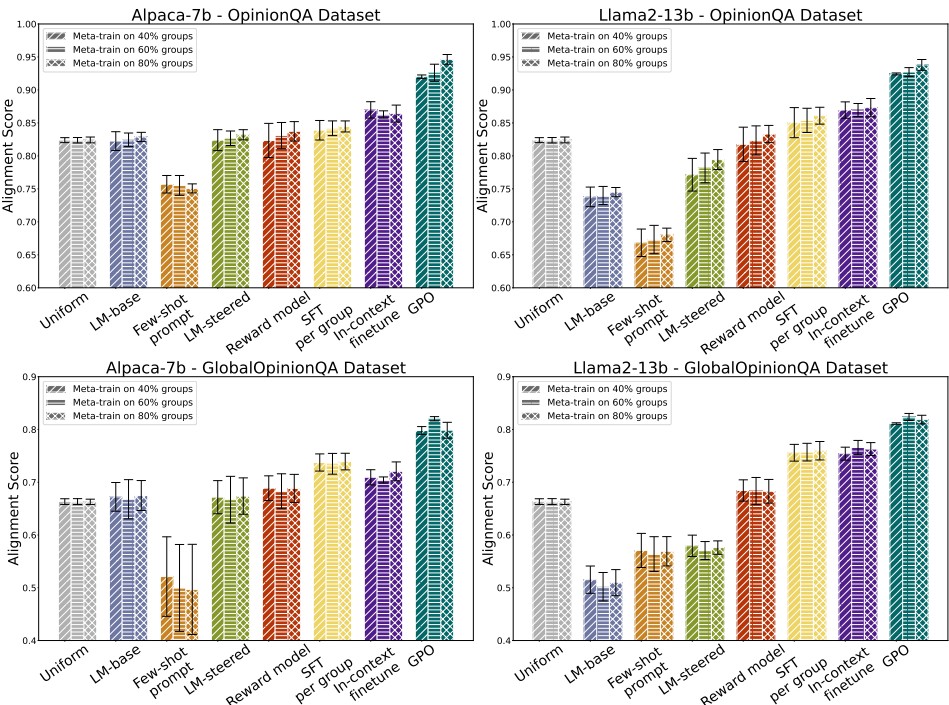

Figure 3: Alignment score comparisons on the OpinionQA dataset and GlobalOpinionQA dataset with Alpaca-7b and Llama2-13b-chat as base models. Results have been averaged across three group split setups and three random seeds, with standard deviations provided.

We report results for the strategy that yields the best alignment as *LM-steered*. Given explicit group information, *Alpaca-7b-steered* displays slightly lower relative gains as compared to *Llama2-13b-steered*. Next, when provided with few-shot group preference context samples, which serve as an implicit method of conveying group information, the LM's alignment performance significantly declines compared to the base language model's performance. We hypothesize this decline might be due to the prompting format being outside the distribution of the language model's training corpus.

For methods involving gradient updates, we maintain a consistent number of context samples across all baselines, which is also the same number of context examples used in *Few-shot prompt*. Specifically, we use 15 samples for Alpaca-7b and 20 for Llama2-13b experiments. With gradient updates, *SFT per-group* brings improvement as compared to other gradient-free steering methods. However, training a *Reward Model* to predict alignment scores from context samples, and subsequently use it to predict preference scores for query examples underperforms SFT methods. This outcome may suggest a risk of overfitting when working with a limited sample size.

GPO achieves notably higher alignment scores on this dataset compared to the baselines for both the Alpaca and Llama2 base models. GPO uses the same number of context samples for adaptation and the test groups are unseen during training. We observed performance increases when a larger number of meta-training groups were used. GPO's closest baseline, the *In-context Finetune* method ranks second, where the LMs are trained to infer from few-shot context samples. On average over the two base models and the three group split settings, GPO achieves a 7.1% increase over the *In-context Finetune*.

Figure 4 qualitatively illustrates the predicted alignment scores from different methods in response to an OpinionQA example concerning climate change concerns across six demographic groups. The first row depicts the ground truth group opinion distribution. Given just 15 context samples, GPO successfully adapts to match the opinion distributions of different groups. For instance, it increases preference for option A when adapted to the group *Hindus*, while the steered LMs do not exhibit correct distribution changes. For example, *Llama2-13b-steered* appears to be biased towards a specific option, overrepresenting it rather than accurately reflecting the distribution of the

targeted group. On the contrary, in demographics with a more balanced distribution like *College graduate/some postgrad*, GPO maintains this balance more consistently. This demonstrates that GPO does not merely adapt to the overall dataset group preferences, but can align to specific groups using limited context.

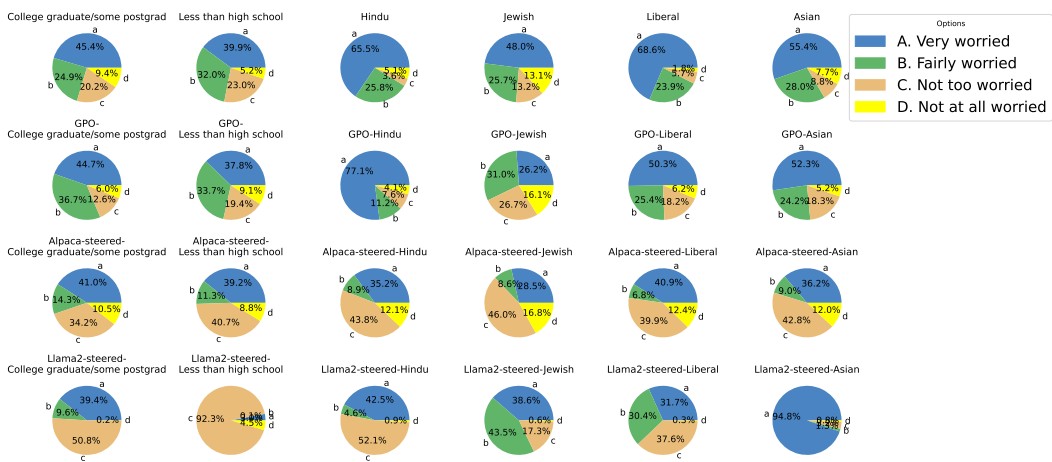

Figure 4: Qualitative comparison of GPO alignment with steered LMs, where each pie chart denotes the preference distribution of the group. Here, GPO uses Alpaca-7b's embedding.

**Adapting to cross-nation groups in GlobalOpinionQA.** The diverse and highly contrasting opinions across nations in GlobalOpinionQA presents a more complex landscape than the OpinionQA dataset. Upon analyzing performance, trends in the GlobalOpinionQA dataset closely followed those observed in the OpinionQA, as depicted in Figure 3. Notably, the alignment score of the Alpaca-7b base model surpasses that of the uniform distribution while Llama2-13b base model shows lower alignment. For Alpaca-7b *LM-base*, this could suggest that the base models might exhibit stronger alignment to certain specific countries and this hypothesis is supported by the increased standard deviation of the Alpaca-7b *LM-base* alignment scores, hinting at varied alignment across different countries, a phenomenon also reported in the dataset (Durmus et al., 2023). Alternatively, this could imply that the base models tend to align more with the dataset's general respondents, which naturally would exceed a uniform distribution. With gradient updates, the *SFT per-group* method here surpasses the alignment performance of steering methods, while the *Reward Model* underperforms SFT methods. The *In-context Finetune* method emerges as the third-best and second-best in terms of alignment for Alpaca-7b and Llama-13b respectively, which showcases enhanced in-context few-shot adaptation post meta-training. However, its training demands are substantially higher; it requires approximately 4.7 times more training time as compared with GPO on an NVIDIA RTX A6000 to achieve the depicted performance. Averaged across both base models and the three group split scenarios, GPO posts a 8.4% improvement over the second-best baseline.

**Scalability with Increasing Context Samples.** We evaluate the scalability of different methods with respect to the size of the in-context examples. Figure 5 demonstrates that for Nigeria in the GlobalOpinionQA dataset, GPO enhances alignment scores with fewer than 10 preference context samples. The performance of *Few-shot Prompt* improves with more examples but plateaus with greater variance. In comparison, *In-context Finetune* exhibits superior adaptability post meta-training than *Few-shot Prompt*, yet its alignment is still suboptimal and the number of group context samples is limited by the context window size of the LM. Both *SFT per-group* and *Reward Model* show incremental improvements with added context samples; however, their sample efficiency is modest. In contrast, GPO adeptly adapts to groups in a sample-efficient manner.

**Adapting to Individual Preferences.** Variations in individual opinions can manifest even within the same demographic groups (Hwang et al., 2023). We align GPO with individual-level preferences.

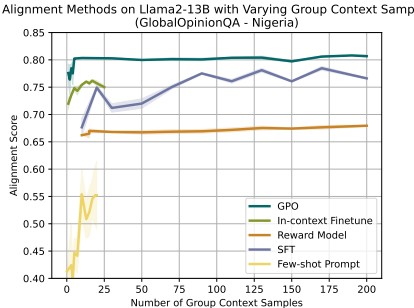

Figure 5: Alignment score of various methods based on Llama2-13B with varying group context sample size. Evaluation conducted on survey questions for Nigeria from the GlobalOpinionQA dataset. The shaded region represents the standard deviation across three different seed results.

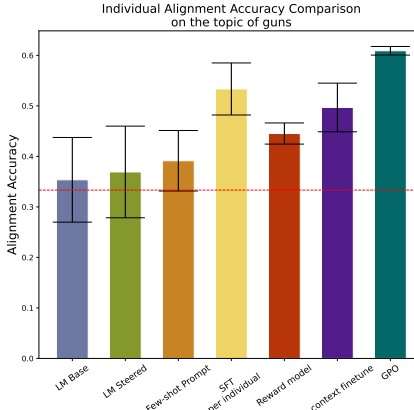
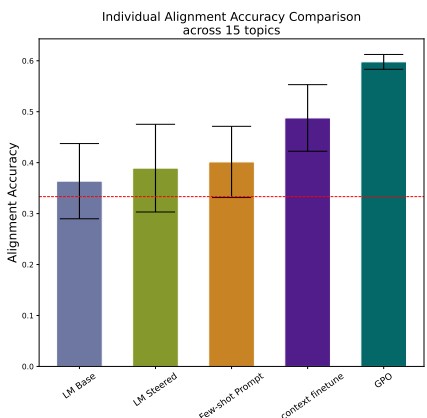

Figure 6: Individual alignment accuracy comparisons from the OpinionQA dataset. **Left:** Individual alignment on the gun topic survey. **Right:** Comprehensive comparison across all 15 topics, showcasing the performance of various methods on diverse subjects. Experiments use Alpaca-7b as the base LM. Both GPO and *In-context finetune* are meta-trained on 40% of individuals and evaluated on the remaining 60%. The horizontal red line represents the average accuracy of a random model.

From the OpinionQA dataset, encompassing 15 surveys across 15 unique topics, we randomly select 100 participants from each survey, along with their responses to 30 topic-related questions. For each individual, 40% questions serve as context samples and 60% for queries. We use Alpaca-7b here as the base model. To steer the LM with individual information, we create individual context from combined demographic variables, such as income, religion, and age, as demonstrated in Appendix Figure 9. Since each individual only selects one option, we calculate alignment accuracy instead by treating the option with the highest predicted preference score as the predicted option. Due to computational constraints, we confined our evaluations of the SFT per-individual and reward model methods to one survey. Since both of them operate on a per-individual basis, the training needed for about a thousand individuals made broader comparisons of the two baselines impractical. In contrast, other baselines, including in-context finetune and GPO, were assessed across all 15 survey topics. Across the full breadth of the 15 topics, GPO consistently exhibited superior performance in adapting to individual preferences relative to other baselines, as depicted in Figure 6.

## 4 CONCLUSION

We introduced, GPO, a novel method for few-shot aligning LLM outputs to both individual and group preferences given little preference data. GPO is trained on a meta-train dataset containing group-wise preference data. During inference, GPO adapts to a new test group, predicting aligned preferences given a few context examples from that group. GPO significantly outperforms prior methods as measured by alignment score for group preference alignment while requiring no gradient updates to the base LLM. We find that GPO is also more sample efficient, improving alignment score significantly more than baseline methods while using fewer samples, and is effective across multiple popular open-source LLMs of various parameter and pre-training dataset scales.

ETHICS STATEMENT

GPO can be used to align models to preferences of diverse interest groups which can provide a more positive, useful, and inclusive experience for end users of LLM applications. We acknowledge that aligning LLMs to the preferences of demographic groups can have malicious applications. For example, making LLMs more capable of producing responses that are more tailored to specific users may be misused to convince or show members of a group how to perform unethical actions. Additionally, GPO's methodology can be used to align a model to a group's preferences even if those preferences are harmful. Biased, offensive, and harmful preferences present in the meta-train or meta-test datasets may be reflected in the outputs of GPO. Future work should investigate methods for aligning LLM outputs to group preferences without amplifying harmful outputs.

ACKNOWLEDGMENTS

This research is supported by a Google Award for Inclusion Research and an Adobe Data Science Award. We want to thank Hritik Bansal for insightful discussions.

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

## A    LIMITATIONS

We highlight a few limitations and directions for future work below:

**Opinion Datasets:** We use datasets containing opinions of various demographic groups to validate GPO. Survey data is imperfect and may not be fully representative of an entire group's population. Additionally, all the datasets that we use in this work are in English. When aligning to groups, the language that is used to collect preference data and during alignment may have a significant effect on alignment metrics, especially if the inputs and outputs are in a different language than the native language of members of a group. Future work should also investigate more challenging few-shot alignment settings, such as adapting to individual creative preferences where there may be much higher variance between group preferences.

**Multiple-choice Format:** Like many previous works, we focus on a multiple-choice format due to the availability of existing datasets and ease of quantitative evaluations. LLMs are capable of producing much more complicated long-form responses, and it is important that alignment methods can be extended to the general long-form response setting. While the GPO framework extends more broadly to different formats of LLM generations, future work should validate the effectiveness of GPO for longer form responses and additional considerations such as group preference feedback representation and evaluation metrics needed to extend to the long-form setting.

**Alignment Objectives:** When aligning LLMs, multiple factors beyond group preference alignment are also very important. Aligning to group preferences may result in worse alignment for other factors including as harmlessness and helpfulness especially if the group preference data includes examples that contradicts these values. Moreover, aligning to group preferences may amplify undesirable behaviors from LLMs including biased or harmful outputs. Future work should study the impact of group alignment on other important alignment factors and methods to reduce regressions for these factors when aligning to group preferences.

**Model Initialization:** Initializing GPO with a pretrained LM transformer backbone might offer advantages in performance. Specifically, leveraging a pretrained backbone could potentially enhance GPO's capacity to encode world knowledge, thereby improving its ability to generalize to OOD examples. Investigating the performance and generalization benefits of this initialization approach could be a promising direction for future work.

## B    DATASET DETAILS

### B.1    OPINIONQA DATASET

This dataset is sourced from Pew American Trends Panel (PewResearch). This dataset's unique structural characteristics: the answer choices in the survey questions are principally ordinal (Santurkar et al., 2023). For instance, options often extend across a spectrum, ranging from categories such as "A great deal," "Fair amount," "Not much," to "Not at all." Traditional divergence metrics, such as the Kullback-Leibler (KL) divergence, are ill-suited for this task, as they fail to encapsulate the ordinal relationships inherent in the answer choices. In this dataset, the ordinal answer choices are mapped to a metric space using corresponding positive integers. For example, a typical mapping in our dataset might look like $\{A : 1, B : 2, \dots, D : 4\}$. Therefore, 1-D Wasserstein Distance metric is used. The alignment score for two opinion distributions $P_1$ and $P_2$ is consequently expressed as:

$$\mathcal{A}(P_1, P_2; Q) = \frac{1}{|Q|} \sum_{q \in Q} \left[ 1 - \frac{\mathcal{WD}(P_1(q), P_2(q))}{N - 1} \right] \qquad (4)$$

Here, $N$ denotes the total number of selectable answer options, excluding the option to refuse. The term $N - 1$ functions as a normalization factor, representing the maximal possible Wasserstein distance in the given metric space. The score is bounded within the interval $[0, 1]$, with a score of 1 indicating perfect alignment between the two distributions.

We employ the dataset as encompassing 22 demographic groups within the US, as outlined in Table 1. Our analysis focuses on 500 contentious questions, characterized by frequent disagreements

among the considered subgroups. These questions are the same ones used in the steerability analysis presented in the OpinionQA dataset (Santurkar et al., 2023).

| Attribute | Demographic Group |
|---|---|
| CREGION | Northeast, South |
| EDUCATION | College graduate/some postgrad, Less than high school |
| GENDER | Male, Female |
| POLIDEOLOGY | Liberal, Conservative, Moderate |
| INCOME | More than $100K+, Less than $30,000 |
| POLPARTY | Democrat, Republican |
| RACE | Black, White, Asian, Hispanic |
| RELIG | Protestant, Jewish, Hindu, Atheist, Muslim |

Table 1: Demographic groups considered in our analysis from the OpinionQA dataset.

## B.2 GLOBALOPINIONQA DATASET

The survey questions in this dataset is sourced from the Pew Research Center's Global Attitudes surveys (PewResearch) and the World Values Survey (Haerpfer et al., 2022). These questions do not generally contain ordinal structures in the options and the ordinal scores are not presented in the datasets. Therefore, we choose to use a different metric for evaluating the alignment in this dataset.

$$\mathcal{A}(P_1, P_2; Q) = \frac{1}{|Q|} \sum_{q \in Q} [1 - \mathcal{JD}(P_1(q), P_2(q))] \tag{5}$$

In this alternate scenario, $\mathcal{JD}$ signifies the Jensen-Shannon Distance following the paper's choice (Durmus et al., 2023).

| Country |
|---|
| Nigeria |
| Egypt |
| India (Current national sample) |
| China |
| Japan |
| Germany |
| France |
| Spain |
| United States |
| Canada |
| Brazil |
| Argentina |
| Australia |
| New Zealand |

Table 2: List of countries considered in our study, from GlobalOpinionQA dataset.

Out of the 138 countries in the original GlobalOpinionQA dataset, we selected a subsample of 14 countries for our study due to computational constraints. We extract all the survey questions that have the target countries' answers. The countries chosen (in Table 2) span several continents to ensure a broad representation in our evaluation. For instance, Nigeria and Egypt cover Africa, while India and China represent Asia. European nations are represented by countries such as Germany, France, and Spain, and the Americas include the United States, Canada, Brazil, and Argentina. Lastly, the Oceania region is represented by Australia and New Zealand.

## C ABLATION ON THE GPO'S TRANSFORMER ARCHITECTURE

We design GPO with inductive biases that satisfy two properties that are important for accurate preference prediction:

**(1) Context Invariance (Property 1.)**: Unlike traditional transformers that utilize positional encodings, GPO omits these encodings to ensure predictions remain unaffected by the sequence or permutation of context preference pairs. However, this loses the pairwise relations between $(x_i, y_i)$. To solve this, we concatenate each pair $(x_i, y_i)$ into a single token to inform the transformer of their pairwise relation. It adopts an alternative masking strategy that differs from the conventional causal mask. This approach enables context pairs to exclusively interact with each other, thereby maintaining focus on relevant information.

**(2) Target Equivalence (Property 2.)**: The masking strategy also makes target points only attend to the context points, which ensures that the targets are only influenced by the context points, not by other targets. This aligns with the principle of conditional independence as stated in Equation 2. In this setup, we don't require the query preference scores to be generated autoregressively, meaning the prediction doesn't depend on previously predicted queries. Therefore, we use a masking strategy where context samples self-attend, while query pairs attend only to context samples, not to other query pairs.

**Property 1. Context Invariance.** A model $p_\theta$ exhibits context invariance if, given any permutation function $\pi$ and any $m \in [1, n-1]$, it satisfies:

$$p_\theta(y_{m+1:n}|x_{m+1:n}, x_{1:m}, y_{1:m}) = p_\theta(y_{m+1:n}|x_{m+1:n}, x_{\pi(1):\pi(m)}, y_{\pi(1):\pi(m)})$$

**Property 2. Target Equivariance.** Model $p_\theta$ demonstrates target equivariance if, for any permutation function $\pi$ and any $m \in [1, n-1]$, the following is true:

$$p_\theta(y_{q,m+1:n}|x_{q,m+1:n}, x_{q,1:m}, y_{q,1:m}) = p_\theta(y_{q,\pi(m+1):\pi(n)}|x_{q,\pi(m+1):\pi(n)}, x_{q,1:m}, y_{q,1:m})$$

To illustrate the effectiveness of these biases, we compare GPO with a standard autoregressive transformer that employs a causal mask, akin to the transformers used in GPT-x series (Radford et al., 2018; 2019; Brown et al., 2020). This basic architecture includes autoregressive generation with the causal mask and uses positional encoding, which we previously omitted to ensure context invariance. Using an autoregressive generation approach violates the target equivalence property since the prediction of each query point relies on previously generated ones. As depicted in Table 3, GPO's inherent inductive biases yield superior alignment performance compared to a traditional transformer. It's noteworthy that in this comparison, we still concatenate the $(x, y)$ pairs into single tokens for the standard transformer, thus preserving the relationship between the viewpoint $x$ and the group preference score.

|  | Meta train on 40% groups | Meta train on 60% groups | Meta train on 80% groups |
|---|---|---|---|
| GPO | **0.798 ± 0.007** | **0.820 ± 0.004** | **0.799 ± 0.015** |
| Transformer | 0.780 ± 0.009 | 0.782 ± 0.004 | 0.772 ± 0.006 |

Table 3: Comparison of the alignment scores of GPO and a standard autoregressive transformer on alignment tasks on GlobalOpinionQA datasets with three group splits and runs are averaged over three seeds. Experiments are conducted on OpinionQA with Alpaca-7b as the base model.

## D    ABLATION ON GETTING EMBEDDINGS FROM THE LLM.

Given that the base LLMs we considered in our experiments were not explicitly trained for text summarizing, we examined three methods to generate the embedding $x$ of the sentence: 1) Using the embedding of the last token as the sentence embedding. 2) Averaging over the embeddings of all tokens in the sentence. 3) Concatenating the embeddings obtained from the previous two methods. As depicted in the table 4, averaging over the token embeddings of the sentence yielded the most effective results, whereas relying solely on the last token embedding proved less adept at capturing sentence-level information.

## E    ABLATION ON ADDING GROUP META-CONTEXT FOR GPO

In the primary experiments, viewpoints $x$ are embedded using an LLM. Notably, in our previous experiments, each $x_i$ does not contain group meta-data about the group's identity or attributes. This

| Embedding Method | Alignment Score |
|---|---|
| Alpaca-7b last token | $0.903 \pm 0.014$ |
| Alpaca-7b average tokens | $\mathbf{0.946 \pm 0.007}$ |
| Alpaca-7b last token + average | $0.942 \pm 0.009$ |

Table 4: Comparison of different embedding methods using Alpaca-7b as the base model on the OpinionQA dataset, with a meta train split of 80%. Results are averaged across three seeds.

ablation study explores the potential performance enhancement that could be achieved by integrating meta-data into GPO. Specifically, the context information $c_g$ is embedded into a vector $z_{ctx}^g$, which is of the same dimension as $x$ as embedded by the same LLM. We examined adding $c_g$ from the three kinds of contextual prompts we study in K. This embedding is then concatenated with each of the $(x, y, z_{ctx})$ pairs, serving as the one input token for GPO. As illustrated in Table 5, incorporating context embeddings into the structure doesn't bolster GPO's performance across the three group split scenarios, instead it performs worse. We hypothesize this outcome arises because GPO, unlike LLMs, lacks comprehensive world knowledge of diverse group attributes, making it challenging to adapt to the meta-data embeddings of unfamiliar groups. Instead, GPO excels in deducing preference distributions based on the available $(x, y)$ context sample pairs.

| | Meta train on 40% groups | Meta train on 60% groups | Meta train on 80% groups |
|---|---|---|---|
| GPO | $\mathbf{0.920 \pm 0.003}$ | $\mathbf{0.926 \pm 0.013}$ | $\mathbf{0.946 \pm 0.007}$ |
| GPO w/ meta-data | $0.900 \pm 0.003$ | $0.916 \pm 0.017$ | $0.926 \pm 0.006$ |

Table 5: Comparison of the alignment scores of GPO with and without meta-data embeddings with three group splits and runs are averaged over three seeds. Experiments are conducted on OpinionQA with Alpaca-7b as the base model.

## F    BASELINES DETAILS

We compare our method against extensive baseline approaches for aligning an LLM's predicted opinion distributions with human groups:

- **Uniform Distribution:** This baseline assumes that all answer options are chosen with equal probability, indicating no preference or bias towards any specific option. For a given question $q \in Q$ with $N$ answer choices, the distribution $P_U(q)$ is represented as: $P_U(q) = \left[\frac{1}{N}, \frac{1}{N}, \ldots, \frac{1}{N}\right]$.

- *LM Base*: The opinion distribution, denoted by $P_\pi$, is derived from a pre-trained LM without any group-specific steering or fine-tuning. For a given question $q \in Q$, the distribution $P_\pi(q)$ generated by the model is extracted from the output probability distribution across the $N$ available answer choices. We first extract the prediction scores for the next token from the LM, focusing on the top-$K$ tokens. We then normalize the values to obtain $P_\pi(q)$. For a token that is missing from the top-$K$ set, we allocate the smallest prediction score in the top-$K$ set. We use $K = 200$ in our experiments.

- *LM Steered*: This baseline gauges the model's adaptability to align with a specific group $g \in G$ when informed of the group information explicitly through the prompt. We use diverse prompting strategies—QA, BIO, and PORTRAY—to convey group information, with examples in Appendix K. The opinion distribution obtained for group $g$ under this steering is expressed as $P_\pi(q; c_g)$, where $c_g$ denotes the context for group $g$.

- *Few-shot Prompt*: Rather than giving the model explicit group information, we input a few examples showing a group's preferences for $m$ context questions, constrained by the LM's context window size. Here the $c_g$ includes the context samples $\{q, r_i, y_i\}_{i=1}^m$. Using this context, the model is prompted to generate a response for a new, unseen question that aligns with the group's opinions. See Figure 8 in the Appendix for examples.

- *SFT per group*: The LM is fine-tuned separately for each group $g$ using a supervised loss. Let $Q_{\text{train}} \subset Q$ denote the subset of $m$ context questions used for training. We create training examples $(q, r)$ by sampling $q$ from $Q_{\text{train}}$ and then sampling responses $r$ with respect to the preference distribution $P_g(q)$. The loss is defined as:

$$L_{\text{SFT}} = -\mathbb{E}_{q \sim Q_{\text{train}}, r \sim P_g(q)} \log p_\psi(r|q) \tag{6}$$

where $\psi$ represents the LM parameters and $p_\psi(r|q)$ denotes the probability of producing the response $r$ given the question $q$. This procedure fine-tunes the LM to maximize the likelihood of the sampled responses that align with the preference distribution of the specific group.

- **Reward Model**: We start with the architecture of the base LM and add a linear MLP head. The augmented model is trained on $m$ context samples to predict the preference scores for the $\{x_i\}_{i=1}^m$ using a mean squared error loss. Then, the model is employed to predict the preference scores for the query questions and softmax is applied to ensure that $\sum_{i=1}^T \hat{y}_{g,q,i} = 1$ for each query $q$.

- **In-Context Finetune**: We investigate whether the LM can be fine-tuned, akin to GPO, to adapt to a distribution of groups using few-shot learning. This would ideally enable improved few-shot in-context adaptation for unseen groups. To this end, we partition the group set $G$ into a meta-train set $G_{\text{train}}$ and a meta-test set $G_{\text{test}}$. During training, each group in $G_{\text{train}}$ serves as a training instance. The training questions for each group are split into context samples and query questions. For a given query question $q$, we supplement it with a few-shot context $c_g$, consisting of $m$ questions paired with the respective ground truth preference scores. This context mirrors the *Few-shot Prompt* strategy with example shown in Appendix 8. For supervision, for each query, we sample responses $r$, aligned with the human preference distribution $P_g(q)$. The LM undergoes fine-tuning using a dataset formed from these context-enhanced samples. The associated loss function is:

$$L_{ICT} = -\mathbb{E}_{g \sim G_{train}, q \sim Q, r \sim P_g(q)} \log p_\psi(r|q, c_g) \tag{7}$$

## G    TRAINING SETTINGS

For all baseline fine-tuning methods, including SFT per group, reward modeling, and in-context fine-tuning that necessitate training the base LM, we employ 8-bit integer quantization and utilize a single Nvidia RTX A6000 GPU with 48GB VRAM. Our parameter search for the learning rate encompassed values {3e-4, 2e-5, 1e-4}. We settled on 1e-4 for the Alpaca baselines and 2e-5 for the Llama2-13B-chat baselines. For both SFT and in-context fine-tuning tasks, our effective batch size was 8, comprised of a batch size of 1 and 8 gradient accumulation steps. In contrast, reward model training had a batch size of 4 with the same gradient accumulation steps. All baseline methodologies were trained with LoRA (with r=12, alpha=32, and a dropout rate of 0.05) with a weight decay of 0.01, utilizing bf16 precision and the AdamW optimizer (Loshchilov & Hutter, 2018). For all methods, We use the validation alignment score for early stopping.

For GPO, the transformer's feedforward dimension was set to 128, with an embedding depth of 4, 4 heads, and 6 layers. We sampled $m$ uniformly from the range [10, 100] as context samples for every training task. We also used a learning rate of 3e-4, coupled with the Adam Optimizer (Kingma & Ba, 2015). More training details can be found in our codebase.

## H    EXTENDING GPO BEYOND MULTIPLE-CHOICE QUESTIONS

The GPO framework presented in the main paper experiments can be extended beyond the multiple choice setting. GPO works for any LLM generation setting where there is some scalar which represents feedback over an LLM response. We present GPO formulations for producing group aligned LLM responses in the long-form generation setting with two common forms of sparse feedback: (1) relative (e.g. is response 1 or response 2 better) and (2) absolute (e.g. rate the response on a scale of 1-7).

**Relative feedback:** each context example includes 2 responses and GPO is trained with a binary classification objective for each example. During inference, the GPO module can be used to perform inference through a modified version of best-of-n sampling where $n$ sample responses are sampled from the base LLM and each of the $\binom{n}{2}$ pairs of responses is inputted to GPO as queries. GPO's output can used to calculate a win rate for each of the $n$ responses and the response with the highest win-rate is chosen as the aligned output response.

**Absolute feedback:** each context example includes 1 prompt and GPO is trained to regress the absolute feedback score. During inference, the GPO module can be used as a reward model in best-of-n sampling to produce a group aligned response.

Since GPO predicts group preference scalars, GPO can be used as a reward model to fine-tune the base LLM with PPO in settings where performing inference with an additional model is not desirable.

## I  ADDITIONAL RELATED WORK

**Alignment via Prompting.**  The conditional nature of LLMs enables them to be conditioned on specific task information or context data and alter their output distribution to respect the conditional information. Various studies have investigated this capability in adapting to groups and personas. Deshpande et al. (2023) observed that prompts like *"Speak like xxx"* could elevate LLM's toxicity levels contingent upon the persona's characteristics. Jiang et al. (2023) used prompts to guide GPT-3.5 to adopt certain personality traits. Beyond explicit persona or group traits, an LLM's behavior can also be influenced by presenting it with few-shot examples from its in-context learning ability (Brown et al., 2020). For example, Hwang et al. (2023) uses the previous opinions of an individual to adapt the LLM to align with the user. The advantages of this strategy include its computational efficiency, eliminating the necessity for gradient updates. However, the few-shot examples are restricted by the model's context size, and designing prompts for effective completion of tasks often requires careful prompt engineering (Lester et al., 2021; Qin & Eisner, 2021; Zhou et al., 2022; Reynolds & McDonell, 2021). Additionally, when steering the LLM to be more representative of a demographics group on nuanced societal questions from survey datasets, especially on nuanced societal matters, research by Santurkar et al. (2023); Durmus et al. (2023) shows that this steerability can be constrained, resulting in limited or no enhancements in model alignment.

**Gradient-based Alignment.**  Another line of alignment work involves adjusting the LM's parameters using a preference dataset through fine-tuning. Methods have been proposed to align LLMs with specific human values, such as helpfulness, harmlessness, and non-toxicity, using RLHF (Glaese et al., 2022; Ouyang et al., 2022; Rafailov et al., 2023; Bai et al., 2022a). This necessitates the modeling of a reward model from human-labeled preference datasets and the use of RL policies, such as PPO (Schulman et al., 2017), to maximize accumulated rewards. This PPO phase poses challenges in terms of computational and memory demands, necessitating the training and storing of large-scale policy, value, reward, and reference models, as well as optimizer states and gradients in GPU memory. Moreover, this process could require complex hyperparameter tuning (Sun et al., 2023; Santacroce et al., 2023). To address this, methods such as Direct Preference Optimization (Rafailov et al., 2023) and Preference Ranking Optimization (Song et al., 2023) have been proposed to directly learn from pairwise or ranking-based preference datasets without reward modeling. However, these methods typically result in specialized models for every alignment task. Adapting to multifaceted, sometimes conflicting group preferences requires fine-tuning distinct models for each subgroup.

## J  QUALITATIVE EXAMPLES OF GPO.

*Warning: This section contains qualitative examples that may be viewed as offensive or harmful.* Here we demonstrate multiple qualitative examples of GPO's predicted group preference versus the language model's steered performance. Here we used only 15 context examples for GPO and the steered LM uses group's meta-data as context.

Q: As you may know, same-sex marriage is now legal in the U.S. Do you think this is a good thing or a bad thing for our society?

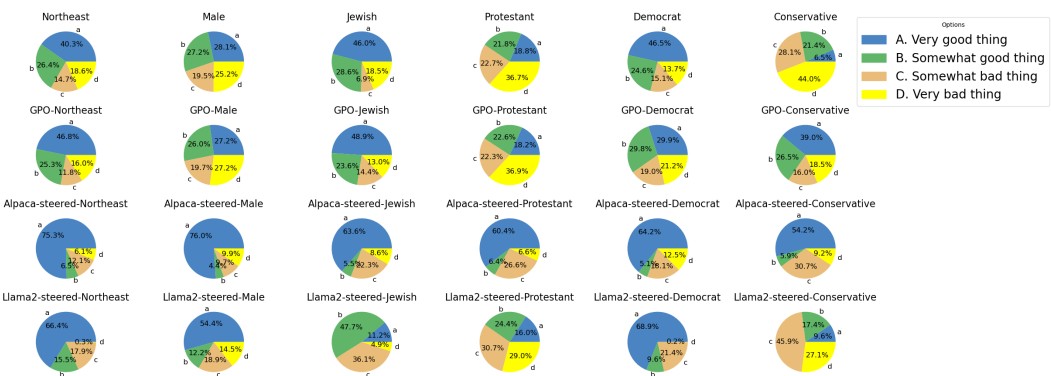

Q: How much, if at all, do you think the following proposals would do to reduce economic inequality in the U.S.?
Increasing taxes on the wealthiest Americans

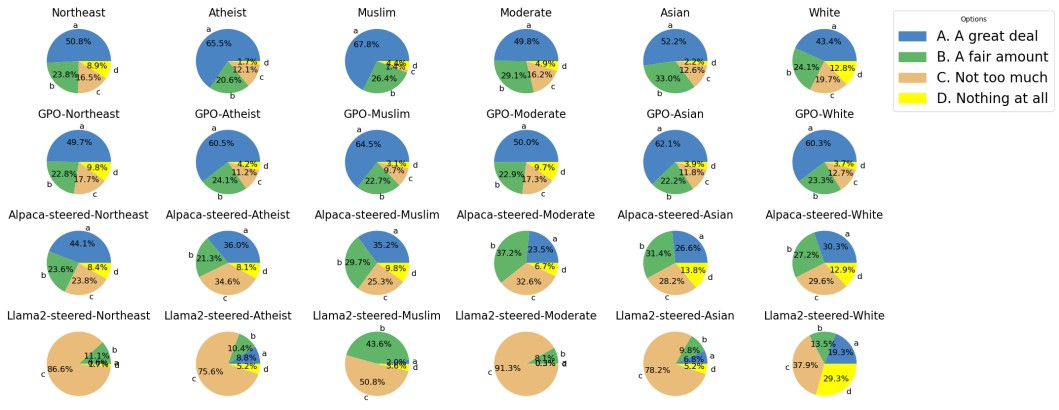

Q: Do you think the number of legal immigrants the U.S. admits should

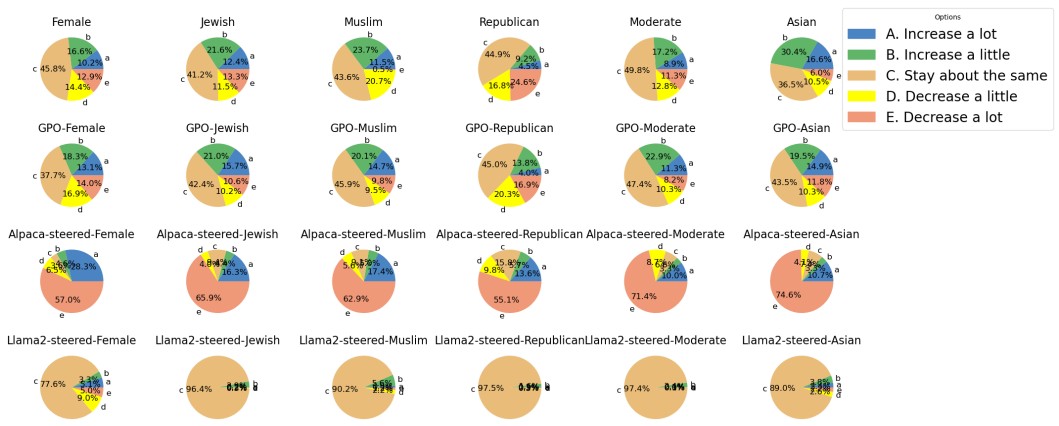

Q: In the future, what kind of an impact do you think major corporations will have in solving the biggest problems facing the country?

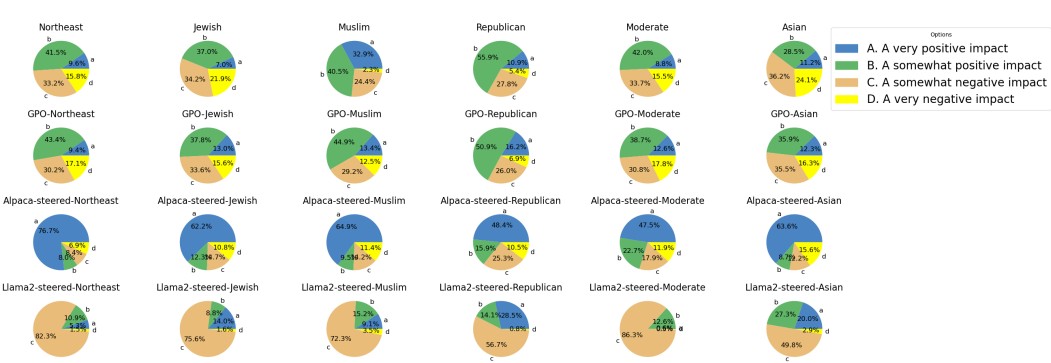

Q: If you were deciding what the federal government should do to improve the quality of life for future generations, what priority would you give to providing high-quality, affordable health care to all Americans?

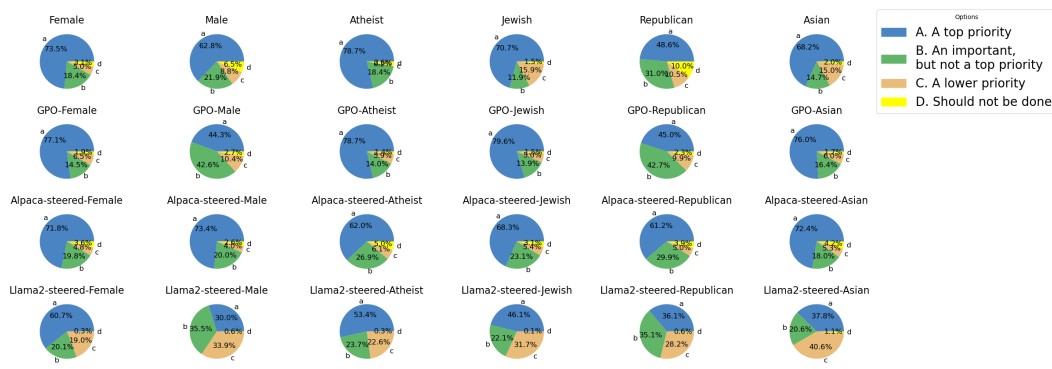

Q: How satisfied are you with the way democracy is working in our country - very satisfied, somewhat satisfied, not too satisfied or not at all satisfied?

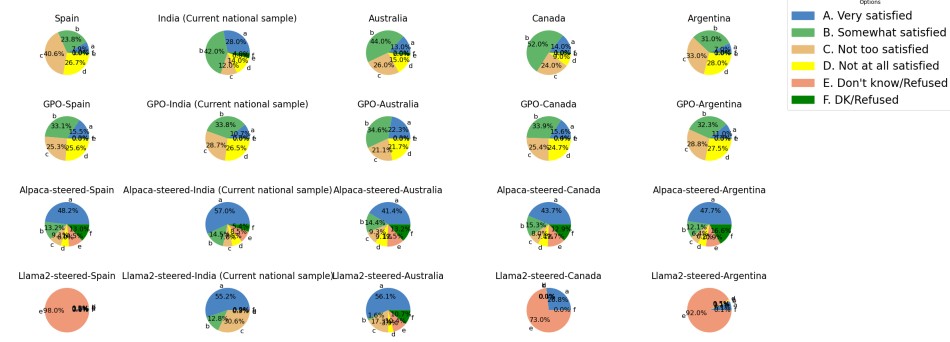

## K  CONTEXTUAL PROMPT EXAMPLES

In this paper, we examine three types of contextual prompts, as delineated in Santurkar et al. (2023). Below, we present examples of the question-answer, biographical, and portrait-based contextual prompts designed for individuals residing in the Northeastern United States.

**Question-Answer Prompt:**
```
Which part of the United States do you currently live in?
Response:  Northeast
```

**Biographical Prompt:**
```
Below, please provide a brief description of the region in
which you currently reside within the United States, followed
by answers to several questions.
Description:  I currently reside in the Northeast.
```

**Portrait-Based Prompt:**
```
Answer the following question as if you currently reside in the
Northeast.
```

Figure 7: Three types of contextual prompts to provide group information.

```
Below is an instruction that describes a task, paired with an
input that provides further context.  Write a response that
appropriately completes the request.

### Instruction:
Given the answer distributions from a specific demographic
group for certain questions in a public opinion survey, answer
the subsequent new question by selecting ONE of the options, as
if you are a member of this identified demographic group:

### Input:

Question:  Question_1
A. Option_1
B. Option_2
C. Option_3
Answer Distribution:
A: 25%, B: 35%, C: 40%

...

Question:  Question_m
A. Option_1
B. Option_2
C. Option_3
Answer Distribution:
A: 35%, B: 25%, C: 40%

Based on the above list of answered questions from a
demographic group, answer the new question by selecting ONE of
the options, as if you are a member of this demographic group:

Question:  Question_m+1
A. Option_1
B. Option_2
C. Option_3

### Response:
```

Figure 8: Few-shot in-context prompt with $n$ context questions in Alpaca prompt format.

```
Below is an instruction that describes a task, paired with an
input that provides further context.  Write a response that
appropriately completes the request.

### Instruction:

Given that you have the following demographics context:
Marital Status:  Married,
Religious attendance:  Roman Catholic,
Region:  Northeast,
Age:  65+,
Sex:  Male,
Education:  Some college or no degree,
Income:  $30,000-$50,000,
Political ideology:  Conservative,
Race:  White,
Answer the following question by picking ONE of the given
options

### Input:

Would you say Germany has done a good or bad job dealing with
the coronavirus outbreak?

Options:
A. Very good
B. Somewhat good
C. Somewhat bad
D. Very bad

### Response:
```

Figure 9: A randomly selected individual contextual prompt examples in Alpaca prompt format.

