# OpenReview forum: "Group Preference Optimization: Few-Shot Alignment of Large Language Models"
_ICLR.cc/2024/Conference — ICLR 2024 poster_

### Official Review · Reviewer_UCgS · 2023-10-28

**Soundness:** 3 good
**Presentation:** 3 good
**Contribution:** 4 excellent
**Rating:** 6
**Confidence:** 4

**Summary:**

The study introduces Group Preference Optimization (GPO), an innovative alignment framework designed to tailor large language models (LLMs) to specific group preferences with minimal data. GPO optimizes the model with reduced data, surpassing current methods like in-context steering and fine-tuning. Experiments on OpinionQA exhibit that GPO effectively adjusted the model's preferences on multiple-choice tasks.

**Strengths:**

1.	The paper commendably focuses on the concept of "Group Preference", introducing an efficient approach for aligning large language models to specific groups.

2.	Efficient fine-tuning of a few-shot learning scenario, blending both in-context and fine-tuning methods, stands out. This approach offers a practical solution in settings where extensive labeled data might not be available.

3.	The empirical evaluations seem robust and thorough. Not only do the results show clear improvements, but the detailed analysis and discussion also provide insightful opinions.

**Weaknesses:**

1.	Lack of clarity in the presentation of the method. For instance, it remains ambiguous which specific parameters are subject to training. Incorporating a detailed algorithm diagram or flowchart would greatly enhance the comprehensibility. Besides, more training details would be better.

2.	The absence of evaluation results concerning the generalization capability of the GPO method. As GPO optimizes the parameters, it's crucial to show whether the model retains performance on general benchmarks after tuning.

**Questions:**

1.	As mentioned, it is ambiguous which part parameters are tunable. Could the authors further clarify the training details?

2.	Could the authors provide examples of generation tasks (QA) to better showcase the model's preference? Demonstrating the model's performance on responses could provide a clearer insight into its practical applications.

---

> ### Author Response · Authors · 2023-11-18
> **response to reviewer UCgS**
>
> Thank you for your valuable comments and detailed insights! We address your questions below.
>
>
> > Q1: Lack of clarity in the presentation of the method. For instance, it remains ambiguous which specific parameters are subject to training. Incorporating a detailed algorithm diagram or flowchart would greatly enhance the comprehensibility. Besides, more training details would be better.
>
> A1: Thanks for pointing out the ambiguity. We added an algorithm box in Appendix A of the revision to illustrate how GPO is trained. GPO trains a transformer module for in-context predicting unseen group preferences based on the few-shot examples provided. And we do not train the base LLM weights. Moreover, the training details and hyperparameters are given in the appendix section G. We hope this clarifies your question. If you have more questions, please let us know.
>
> > Q2: The absence of evaluation results concerning the generalization capability of the GPO method. As GPO optimizes the parameters, it's crucial to show whether the model retains performance on general benchmarks after tuning.
>
> A2: GPO does not alter the base LLM's parameters. Instead, it augments the LLM with an additional module trained to predict group preferences based on the few-shot examples provided. This approach allows the LLM to retain its broad language capabilities while aligning its outputs to specific group preferences​​ in a few-shot manner.
>
> > Q3: Could the authors provide examples of generation tasks (QA) to better showcase the model's preference? Demonstrating the model's performance on responses could provide a clearer insight into its practical applications.
>
> A3: We agree such datasets could provide valuable insights into GPO's capabilities. However,
> * We want to highlight that GPO is not inherently restricted to multiple-choice datasets. For open-ended long-form QA, GPO could predict raw preference scores and guide the LLM to align with group distributions, without requiring the scores to sum to 1.
>
> * To the best of our knowledge, we are unaware of suitable labeled long-form QA group preference datasets for real world settings which are diverse, nuanced, and seperated for diverse demographic groups that would allow benchmarking.
>
> * Collecting these dataset would be time consuming and is outside of this work’s scope. Additionally, evaluating the long-form generations would require human judgement which is expensive in our academic research setting. For example, gathering diverse demographic groups labellers to evaluate generated responses and aggregate their reponses as group preference distributions would be impractical for academic research.
> * That said, we have made our best efforts to utilize the most comprehensive opinion datasets available for our current evaluations on multiple-choice QA. We chose OpinionQA and GlobalOpinionQA, which is of high community interest given their growing adoption, with 60+ GitHub stars recently for the former and 300+ monthly downloads of the latter. Additionally, their survey QA format enables efficient, accurate analysis without costly human evaluation.
>
> If you have suggestions on such long-form QA datasets, please let us know. We would not hesitate to test GPO on them.

---

> > ### Author Response · Authors · 2023-11-20
> > **reminder**
> >
> > Thank you again for your insightful feedback on our paper! We've carefully considered your comments and revised our paper to address the questions you raised. As the discussion phase is drawing to a close, we hope you've had a chance to review our response. Considering the time constraint, are there any other questions or concerns you'd like us to address/clarify? Your support is greatly appreciated and we sincerely value your guidance in enhancing the quality of our work.

---

> > ### Comment · Reviewer_UCgS · 2023-11-21
> >
> > Thank you to the authors for the response! While better benchmarks may not currently exist, I believe the current evaluation datasets are somewhat limited, which might affect the broader impact of your work. I will maintain the current score.

---

### Official Review · Reviewer_DqhU · 2023-11-01

**Soundness:** 3 good
**Presentation:** 4 excellent
**Contribution:** 2 fair
**Rating:** 5
**Confidence:** 4

**Summary:**

This work proposes a method to alignment LLM to group preferences so that a LLM grounded on several shots of preference examples can better predict the group preference for the unseen questions.

**Strengths:**

1. The problem this work proposes to solve is kind of new and unique.
2. The baselines are quite comprehensive and the proposed method significantly outperforms all of them, which validates the superiority of this method.

**Weaknesses:**

1. I am not sure about the significance and broadness of the problem this work tries to solve. Alignment of LLMs to group preferences sounds important, however, the evaluation datasets used by this work look quite specific and narrow and I don't think it is of interest to a broad range of research community.
2. The proposed method look quite trivial and standard, which is a in-context fine-tuning method. There are some changes in the method details but those details are a bit hard to understand, which I will pose some questions on next.

**Questions:**

1. In the second last paragraph of page 4, it is said that "In particular, we discard the positional encodings commonly found in standard transformer architectures". Wouldn't the removal of positional encodings significantly deteriorate the performance of those pre-trained and fine-tuned LLMs?
2. Still in the second last paragraph of page 4, it is said that "we employ a masking strategy where the context pairs can self-attend to each other". Could you elaborate such a masking strategy?

---

> ### Author Response · Authors · 2023-11-18
> **Response to reviewer DqhU**
>
> Thank you for your valuable comments and detailed insights! We address your questions below.
>
> > Q1:I am not sure about the significance and broadness of the problem this work tries to solve. Alignment of LLMs to group preferences sounds important, however, the evaluation datasets used by this work look quite specific and narrow and I don't think it is of interest to a broad range of research community.
>
> A1:Thanks for acknowledging the importance of steering LLMs to group preferences. We think it is an important task as current popular alignment methods such as prompting, in-context steering, and SFT do not work well for it.
>
> Regarding the evaluation datasets:
> * We have exerted our best effort to utilize the most comprehensive and relevant opinion datasets currently available for assessing LLM opinion distributions. Additionally, there are currently no long-form QA datasets with labeled ratings collected for diverse demographics groups. We chose OpinionQA and Anthropic's GlobalOpinionQA for their growing use and recognition in the field. For instance, OpinionQA has garnered over 60 stars on github in about half a year. Similarly, GlobalOpinionQA has seen over 300 downloads in the past month, indicating high community interest in LLM opinion distribution studies.
>
> * Our selection was also larged guided by the practical constraints of academic research, where datasets that allow for easy verification and no costly human evaluation are preferred. This motivation aligns with OpinionQA (as stated in [1] section 2.1) and GlobalOpinionQA’s motivations too, which states that the use of survey QA data enables capturing diverse and nuanced public interest topics and also enables efficient and accurate evaluation.
>
> * Additionally, in contrast to prior work that treats human alignment as a single quantity such as safety or helpfulness, our work conceptualizes human alignment as an inherently subjective measure that depends on who it is evaluated against; to the best of our knowledge, **we are among the first to align LLMs to a group preference distribution rather than a single preference or group consensus.**
>
> [1] Whose Opinions Do Language Models Reflect?  https://arxiv.org/abs/2303.17548
>
> > Q2: The proposed method look quite trivial and standard, which is a in-context fine-tuning method. There are some changes in the method details but those details are a bit hard to understand, which I will pose some questions on next.
>
> A2:
> * We would like to clarify that GPO is not an in-context fine-tuning method, it does not alter the base LLM's parameters. Instead, it augments the LLM with an additional transformer module trained to predict unseen group preferences based on the few-shot examples provided. Although GPO does not modifying the weights of the LLM itself, it can guide the LLM to produce sample outputs that are attuned to the desired group preferences. This allows the LLM to retain its broad language capabilities while aligning its outputs to specific group preferences​​ in a few-shot manner.
> * In section 3 (Page 6), the in-context finetuning method is actually one of our baselines that performs worse than GPO.
>
> >Q3: ... it is said that "In particular, we discard the positional encodings commonly found in standard transformer architectures". Wouldn't the removal of positional encodings significantly deteriorate the performance of those pre-trained and fine-tuned LLMs?
>
> A3: We discard positional encoding for GPO but not the base LLM. We do this to achieve the property of context invariant, which means the prediction of preference for new questions should not depend on the ordering of the context questions. In the appendix Table 3 we submitted, we compared GPO with a standard autoregressive transformer that employs standard causal mask and positional encodings. And GPO’s inherent inductive biases yield superior alignment performance compared to a traditional transformer.
>
> > Q4: in the second last paragraph of page 4, it is said that "we employ a masking strategy where the context pairs can self-attend to each other". Could you elaborate such a masking strategy?
>
> A4: Thank you for pointing out the potential confusion in our writing. We will provide more details on this in the revision. The masking strategy is as follows:
>
> Our input to the transformer is a sequence of context pairs followed by padded target pairs as in figure 2. The mask allows:
> 1) The context pairs attend to each other, rather than using a causal mask that restricts each token to only attend to previous tokens. We use this because the order of the context pairs should not influence predictions for target points - it should be permutation invariant.
> 2) The padded target pairs to only attend to the context pairs, and not to other targets. This restriction aligns with the conditional independence assumption in Equation 2, ensuring each target pair is predicted solely based on the context.
>
>
> If you have other questions, please let us know.

---

> ### Author Response · Authors · 2023-11-20
> **reminder**
>
> Thank you again for your insightful feedback on our paper! We've carefully considered your comments and revised our paper to address the questions you raised. As the discussion phase is drawing to a close, we hope you've had a chance to review our response. Considering the time constraint, are there any other questions or concerns you'd like us to address/clarify? Your support is greatly appreciated and we sincerely value your guidance in enhancing the quality of our work.

---

> > ### Author Response · Authors · 2023-11-21
> > **reminder**
> >
> > Thank you again for your insightful feedback on our paper! We've carefully considered your comments and revised our paper to address the questions you raised. As the discussion phase is drawing to a close tomorrow, are there any other questions or concerns you'd like us to address/clarify? Your support is greatly appreciated and we sincerely value your guidance in enhancing the quality of our work.

---

> ### Author Response · Authors · 2023-11-22
> **last day reminder**
>
> Since we have reached the end of the rebuttal period, we would like to reiterate and summarize our response earlier.
>
> > **Q2**: The proposed method look quite trivial and standard, which is a in-context fine-tuning method. There are some changes in the method details but those details are a bit hard to understand, which I will pose some questions on next.
>
> * We **added an algorithm box in Appendix A** of the revision to illustrate how GPO is trained. GPO trains a transformer module for in-context predicting unseen group preferences based on the few-shot examples provided. And we do not train the base LLM weights. Moreover, the training details and hyperparameters are given in the appendix section G.
>
>
> * GPO is not an in-context fintuning method. In section 3 (Page 6), the in-context finetuning method is actually one of our baselines that performs worse than GPO. Another advantage is that GPO operates in the latent space and can incorporate more in-context examples, while "in-context fine-tuning" takes raw text tokens as input and is constrained by the context window size. By working in the latent space, GPO can leverage more few-shot examples to better adapt to the preferences of unseen groups at test time.
>
> * To our knowledge, GPO is among the first to do **in-context preference learning** for group alignment - we train a single model that can align to unseen groups at test time without any gradient updates: $p(y_{m+1:n} | x_{1:n}, y_{1:m})$ only with $m$ context pairs. This makes GPO efficient for few-shot adaptation to the nuanced distribution of preferences across groups which can be utilized to guide LLM as a group opinion simulator for downstream tasks.
>
> > **Q3**: ... it is said that "In particular, we discard the positional encodings commonly found in standard transformer architectures". Wouldn't the removal of positional encodings significantly deteriorate the performance of those pre-trained and fine-tuned LLMs?
> >
> > **Q4**: in the second last paragraph of page 4, it is said that "we employ a masking strategy where the context pairs can self-attend to each other". Could you elaborate such a masking strategy?
>
> Regarding the above concerns about positional encodings and masking strategy, we have **added a more detailed description in Appendix C** in the revision to formally describe the two properties integrated in GPO:
>
> 1. **Context invariance**: GPO removes positional encodings, ensuring that predictions are not influenced by the order or permutation of context pairs. It employs a masking strategy other than the causal mask in traditional transformers, such that the context pairs can and only attend to all the other context pairs.
>
> 2. **Target equivalence**: The masking strategy also ensures the targets attend to context points but not other targets, adhering to the conditional independence in Equation 2.
>
> We hope this addresses your concerns. Given *today is the last day for the discussion period*, please let us know if you have more questions. Your support is greatly appreciated and we sincerely value your guidance in enhancing the quality of our work.

---

### Official Review · Reviewer_dRt2 · 2023-11-08

**Soundness:** 3 good
**Presentation:** 2 fair
**Contribution:** 2 fair
**Rating:** 6
**Confidence:** 4

**Summary:**

This paper proposes a framework to tackle the preference prediction problem, that is given a question, predict a distribution over all possible answers provided by the format of multiple choice question. The authors propose to view the question as a semi-supervised prediction problem and use LLM to augment the input data x into a (x,r) pair. The final prediction is done by training a shallow Transformer model over the augmented.

**Strengths:**

The paper proposes a novel idea to augment the data. The final analysis shows that direct tuning on LLM would not also obtain the best qualities on few-shot datasets.

**Weaknesses:**

While the paper presents a new infra on the problem of preference distribution, the alignment method does not look too much different than a normal semi-supervised framework and there are some caveats in the experiment design and baseline choice to fully justify the acceptance of this submission.

* For the baseline, the authors seem to fail to include one direct method.
  * A simple Transformer model that purely learns from the (q, y) pair and uses them in a semi-supervised fashion as a sequence of inputs. As is pointed out by the author, the Reward Model baseline is underperforming a lot of the other baselines, it would really make sense to add a comparable-sized baseline as in the author’s proposed method to rule out the possibility that overfitting is the only cause of inferior baselines w/o really relying on LLM.
* The PeftConfig configuration seems not consistent with the description of the Reward Model baseline in the paper. The authors argue to use a linear MLP head for the Reward Model baseline. However, in the code, the authors used a LoraConfig which should by default fine-tune every layer of LLM, and might be the major reason for the underperformed score of this baseline.
Also, it makes more sense to use cross-entropy for the loss function instead of MSE as the final output is a distribution.
* The terminology of alignment seems a bit too abused and distracting in this setting. In this work, the authors only tried to learn a separate Transformer architecture that operates upon the output of LLM while the LLM itself does not enjoy any new abilities in its parameters based on the modifications.
* It would also be an interesting point in this work to justify the reason for the output of (r) as input in this work (through ablation studies). My current understanding is that the sampled response would be used as an anchor point for the training of the proposed method, however, it would lead to a natural question: without r, will the model underperform a lot, meaning that LLM would also be the important ingredient? Also, in this case, it seems that the qualities of the response from the model would also seem to be quite important, and it would make more sense to replace it with random strings etc to further make the study more coherent.

**Questions:**

It would be great to see the authors compare with the very basic baseline, perform ablation studies, and fix some config settings to make the paper more complete.

---

> ### Author Response · Authors · 2023-11-18
> **response to reviewer dRt2 (1/n)**
>
> Thank you for the detailed and constructive reviews and suggestions for experiments, which have helped improve both the clarity and the rigor of our paper. We addressed your feedback below:
> > Q1. While the paper presents a new infra on the problem of preference distribution, the alignment method does not look too much different than a normal semi-supervised framework
>
> A1: We want to clarify that GPO is not a semi-supervised framework, but rather a few-shot learning framework that is trained on fully labeled preference dataset of (q, r, y), where q is a query question, r is the response, and y is the group’s preference to this response given the q.
>
> GPO’s transformer module learns to infer preference y for a viewpoint (concatenation of q and r), for unseen groups at test time, which is both more accurate and sample efficient than SFT or prompting methods. For better clarity, we have included a detailed algorithmic box for GPO’s training pipeline in Appendix A of the revision paper.
>
> > Q2. For the baseline, the authors seem to fail to include one direct method. A simple Transformer model that purely learns from the (q, y) pair and uses them in a semi-supervised fashion as a sequence of inputs
>
> A2: Thank you for highlighting the potential oversight in our baselines. We'd like to address this in two parts based on our understanding of your suggestion:
> * Based on our problem setup, we think the q you referred to is actually x (the concatenation of q and r), because the preference y is associated with x rather than q alone. Then the concept of inputting the (x, y) pairs directly to a transformer model closely aligns with our 'In-Context Finetune' baseline. In this baseline, n context pairs (x,y) are indeed fed sequentially as inputs to the LLM to predict the preference y, and this method performs worse than GPO in our experiments. However, we are not using GPO nor the baseline in a semi-supervised fashion but rather train them on labeled (x, y) pairs.
> * Secondly, we have also included a baseline that aligns with your suggestion of a simple transformer model in appendix Table 3 we submitted. We compared GPO with an equally-sized standard autoregressive transformer that employs standard causal mask and positional encodings and also takes in (x,y) pairs as sequence inputs too. In contrast, GPO has a different masking strategy and removed positional encoding to adopt inductive biases that lead to two properties. One is context permutation invariance, where the prediction of queries does not depend on the ordering of the context pairs; and the other property is target equivariance, which requires that whenever we permute the target inputs, the predictions are permuted accordingly. These inductive biases are important for our problem setup. And GPO’s inherent inductive biases yield superior alignment performance compared to a traditional transformer as shown in the table below.
>
> If our interpretations do not fully address your baseline suggestion, please give us further clarification and we will provide additional baseline experiments.
>
> > Q3: The terminology of alignment seems a bit too abused and distracting in this setting. In this work, the authors only tried to learn a separate Transformer architecture that operates upon the output of LLM while the LLM itself does not enjoy any new abilities in its parameters based on the modifications.
>
> A3: We use "alignment" to mean aligning the output distributions of the generative model - a combination of LLM and GPO - to group preferences. Although GPO does not modify the weights of the LLM itself, it can guide the LLM to produce sample outputs that are attuned to the desired preferences.
>
> This terminology is also in line with prior works for alignment such as [1] and [2], where alignment also refers to tuning sample outputs rather than altering LLM parameters. In these works, alignment is achieved using techniques like reward models and Best-of-N sampling over the pretrained LLM, without updating the base model itself.
> Additionally, in contrast to prior work that treats human alignment as a single quantity such as safety or helpfulness, our work conceptualizes human alignment as an inherently subjective measure that depends on who it is evaluated against; to the best of our knowledge, we are among **the first to align LLMs to a group preference distribution rather than a single preference or group consensus.**
>
> [1] Let’s verify step by step. https://arxiv.org/abs/2305.20050
>
> [2] Scaling Laws for Reward Model Overoptimization  https://arxiv.org/pdf/2210.10760

---

> ### Author Response · Authors · 2023-11-18
> **response to reviewer dRt2 (2/n)**
>
> > Q4:  "the Reward Model baseline is underperforming a lot, it would really make sense to add a comparable-sized baseline as in the author’s proposed method to rule out the possibility that overfitting is the only cause of inferior baselines w/o really relying on LLM. The PeftConfig configuration seems not consistent with the description of the Reward Model baseline..."
>
> A4: Our reward models are initialized using the HuggingFace AutoModelForSequenceClassification which adds an additional linear layer to the LLM. When this model is converted to Peft model, the Peft library ensures that only the LoRA adapters and the additional linear layer are trained since our task_type is set to SEQ_CLS in our PeftConfig and we are using an AutoModelForSequenceClassification model. In the description of reward model baseline in section 3, when we state “we train a per-group reward model by adding a linear MLP head on a base LLM and train it on m context samples”, “it” refers to all of the trainable parameters of the entire Peft model. We have updated the reward model baseline description to more clearly reflect the reward model training setup and apologize for the confusion.
>
> Related works including InstructGPT[1] typically train a reward model by fine-tuning all parameters of a reward model(including the base LLM). Considering that we are training a very small subset of the base LLM’s parameters with LoRA (not the full 7B), compared to prior works which train all 6B+ parameters for reward model baselines, it is unlikely that our reward models are overfitting majorly due to large model size. The number of trainable parameters for the reward linear layer, our original LoRA training and GPO’s transformer module are shown in the table below.
>
> | Model    | Trainable parameter count |
> |---------------------|---------------------------|
> | GPO                 | 1188354                   |
> | Linear reward layer | 4096                      |
> | LLM + Linear layer, with LoRA           | 6295552                   |
>
> Nevernethess, we conducted an experiment to rule out the overfitting possibility due to trainable parameters, and the results are shown below in the table, which shows no improvement for training only the top linear classification layer. We believe the reward model’s inability still lies in the difficulty of learning a high-quality reward function from a small number of training examples for every group.
> |            Reward model setup              | OpinionQA         | GlobalOpinionQA  |
> |--------------------------|-------------------|------------------|
> | Train only top linear layer | 0.824 ± 0.014      | 0.675 ± 0.028    |
> | Orig setup: train both linear and LLM layers with LoRA | 0.831 ± 0.019      | 0.683 ± 0.033    |
>
>
> [1] InstructGPT: https://arxiv.org/pdf/2203.02155.pdf
>
>
> > Q5: Also, it makes more sense to use cross-entropy for the loss function instead of MSE as the final output is a distribution.
>
>
> A5: We chose to use MSE as the loss function because the reward model takes a single input viewpoint (x) and outputs a scalar target value (y) rather than a probability distribution. However, you raise a good point that cross-entropy loss may better match the probabilistic nature of our tasks. Given your suggestion, we experimented with CE loss on the two datasets as shown in the results table. Empirically, the two losses perform similarly, but CE has the advantage that the loss converges faster than MSE loss.
> |          | OpinionQA      | GlobalOpinionQA |
> |----------|----------------|-----------------|
> | CE loss  | 0.820 ± 0.032  | 0.680 ± 0.029   |
> | MSE loss | 0.831 ± 0.019  | 0.683 ± 0.033   |
>
> > Q6: ...justify the reason for the output of (r) as input in this work (through ablation studies) ... a natural question: without r, will the model underperform a lot, meaning that LLM would also be the important ingredient? Also it seems that the qualities of the response from the model would also seem to be quite important, and it would make more sense to replace it with random strings etc to further make the study more coherent.
>
> A6: We think there may be some confusion about the role of the response in our setup. To clarify, the response r is essential in our problem formulation. Our goal is to model preferences y over query-response pairs (q, r), where r provides the crucial context to ground each opinion. Without r, the score y would be assigned to the question q alone. Thus, r is critical for accurately reflecting the nuanced "opinion" we aim to infer group preferences over.
>
> However, we appreciate your suggestion of an interesting ablation and have conducted this experiment by giving a random string response to every query. As shown in the table below, the performance degraded substantially without the response.
>
> | GPO setup  | OpinionQA          |
> |---------------------------|--------------------|
> | Random string response    | 0.7909 ± 0.009  |
> | Actual response           | 0.9201 ± 0.0026    |

---

> ### Author Response · Authors · 2023-11-20
> **Reminder**
>
> Thank you again for your insightful feedback on our paper. We've carefully considered your comments and conducted new experiments to address the questions you raised. We'd like to note that the discussion phase is drawing to a close, and we hope you've had a chance to review our response. Considering the time constraint, are there any other questions or concerns you'd like us to address? Your support is greatly appreciated and we sincerely value your guidance in enhancing the quality of our work.

---

> > ### Comment · Reviewer_dRt2 · 2023-11-21
> >
> > I would like to thank the authors for the detailed response!
> >
> > I think the authors did a good job in answering my questions, which majorly arose from the unclear setup statement about the role of value r (a default value that is provided as part of the input).
> >
> > The setup of the in-context learning becomes clearer after the Algorithm Box A.
> >
> > For the explanation on y, could the authors explain a bit more about why it’s a scaler? Based on Figure 1, I suppose the problem setup is to predict y given a pool of ((q, r), y) tuples where y is a distribution over the options from q. And authors also argue that the work is to “align LLMs to a group preference distribution”.
> >
> > I also want to ask for clarification one more time on the baseline of Transformer vs GPO. For these two methods, is in-context learning the major difference? i.e. for Transformer, only a ((q,r), y) pair is fed each time to the model to train.
> >
> > For the novelty part, I appreciate the authors’ response in stressing the major difference between their work and the existing ones on preference distribution and group consensus. I believe it could be better and clearer if the authors can explain more about the differences between their work vs existing literature in the machine learning jargon. That is, for the existing work of group consensus, did the existing works only show scalers, what is their predicting target and technique that the new method could differ most from?
> >
> > One final suggestion is that the clarity of the draft can be clearer if the authors could move the Algorithm Box and the comparing baseline of Table 3 to the main body.
> >
> > I would be happy to raise my point to weakly accept if the authors could add some comments on my questions.
> >
> > Minor: Why the Meta train on 60% would have the highest qualities? In Table 3.

---

> > > ### Author Response · Authors · 2023-11-21
> > > **response to reviewer dRt2 (3/n)**
> > >
> > > Thank you for the prompt response and further insightful suggestions! Here are our responses to help clarify further:
> > >
> > > >Question 1: 'could the authors explain a bit more about why y it’s a scaler? ...And authors also argue that the work is to “align LLMs to a group preference distribution”'.
> > >
> > > y is a scalar representing the preference score for a particular response option given a query q. For a question q with n response options, we construct n (q,r_i) pairs and get n scalar y_i values. We then apply a softmax to convert these raw scalar values into a probability distribution over the n options to get a preference distribution.
> > >
> > > >Question 2: …on the baseline of Transformer vs GPO. For these two methods, is in-context learning the major difference? i.e. for Transformer, only a ((q,r), y) pair is fed each time to the model to train.
> > >
> > > “for Transformer, only a ((q,r), y) pair is fed each time to the model to train.” The baseline you referred to would mix all the preference data from different groups during training, and the model will not have any sense of group identity and likely predict outputs close to the mean preference distribution averaged over all groups.
> > >
> > > In our experiment, both the GPO and the standard transformer baseline employ in-context learning. They are fed ((q,r), y) pairs as context points and are tasked with predicting padded target (q,r) pairs. GPO differs through specific inductive biases that enable two properties for accurately predicting new (q,r) preference scores:
> > >
> > > * Context invariance:
> > > GPO removes positional encodings, ensuring that predictions are not influenced by the order or permutation of context pairs.
> > > It employs a masking strategy other than the causal mask in traditional transformers, such that the context pairs can and only attend to all the other context pairs.
> > >
> > > * Target equivalence:
> > > The masking strategy also ensures the targets attend to context points but not other targets, adhering to the conditional independence in Equation 2.
> > >
> > > A standard causal autoregressive transformer would violate these properties.
> > >
> > > >Question 3: …explain more about the differences between their work vs existing literature in the machine learning jargon, for the existing work of group consensus, did the existing works only show scalers, what is their predicting target and technique that the new method could differ most from?
> > >
> > > Prior group alignment work targets unidimensional preferences such as safety and helpfulness [4], or the most prototypical traits of a group or individuals [2,3] and evaluates through human eval or APIs. Although they do not explicitly predict a single scalar y, the concept is implicitly applied in their evaluation and methods as follows:
> > >
> > >
> > > (1) Find a better prompt to steer the LLM to act in accordance with a group: $\pi(prompt)$
> > >
> > > (2) learn a single preference by SFT on the specific group dataset with most preferred query and response pairs.
> > >
> > > (3) or learn reward models that take in a prompt and response and output a scalar $f(q, r)$ and use Best-of-N to align LLM output to a group consensus that maximizes agreement across all people in a group [1].
> > >
> > > In contrast, GPO aims to align the LLM to the opinion distribution of groups. Nevertheless, the above consensus alignment is also achievable with GPO as it is equivalent to taking the argmax of the predicted preference distribution from GPO. Additionally, prior alignment methods, like SFT and reward models, require training a separate model for each group's preference data.
> > >
> > > GPO is the first method to do **in-context preference learning** for group alignment - we train a single model that can align to unseen groups at test time without any gradient updates: $p(y_{m+1:n} | x_{1:n}, y_{1:m})$ only with m context pairs. This makes GPO efficient for few-shot adaptation to the nuanced distribution of preferences across groups which can be utilized to guide LLM as a group opinion simulator for downstream tasks.
> > >
> > > [1] https://arxiv.org/pdf/2211.15006
> > >
> > > [2] https://arxiv.org/pdf/2305.02547
> > >
> > > [3] https://arxiv.org/abs/2305.14929
> > >
> > > [4] https://arxiv.org/pdf/2203.02155
> > >
> > > >Question 4: Minor: Why the Meta train on 60% would have the highest qualities? In Table 3.
> > >
> > > We agree generally larger meta-training sets enhance meta-learning performance. However, in Table 3 with the GlobalQAdataset covering 14 countries, setting meta-training at 80% limits testing to just 2 countries. We hypothesize that the observed differences in performance are attributable to the varying intrinsic alignment difficulties that are present among these groups.
> > >
> > > >…the draft can be clearer if the authors could move the Algorithm Box and the comparing baseline of Table 3 to the main body.
> > >
> > > Thanks for the suggestions. We agree with the feedback and will move these components into the main paper in the revision.
> > >
> > >
> > > Please let us know if you have any other questions! We appreciate you taking the time to clarify our method and paper, and the constructive feedback and engagement.

---

> > > > ### Comment · Reviewer_dRt2 · 2023-11-22
> > > >
> > > > Thanks for the detailed explanation.
> > > >
> > > > I think the authors answered all my questions clearly, whereas the Transformer baseline may still seem a bit naive. I guess a better way would be to train for each category a separate model so that whenever the group identity is told, we could use the corresponding Transformer to make a prediction. In this way, the model will not mix up the preference data.
> > > >
> > > > One more note here, which I think was originally confusing is about the scaler of y. The authors answered my question that y is used to pair with the response so that it's a scaler. Afterwards, a softmax score can be calculated. It still seems an unnatural setup: why not just compile every y and r together so that the model could directly predict the distribution all at once while resorting to making individual scores? This method can also result in a scaling issue as the number before the softmax has any associated meaning and can be scaled by the same constant.
> > > >
> > > > Overall, I would be still happy to increase my score.

---

### Meta-Review · Area_Chair_1Eoq · 2023-12-08

**Metareview:**

The submission introduces an approach called group preference optimization (GPO) which trains an auxiliary Transformer network to represent a group's preference on a new input given examples of their preferences on previous inputs in an in-context manner. The approach is applied to Alpaca 7B and an instruction-tuned LLaMA and evaluated on OpinionQA and GlobalOpinionQA and is shown to outperform an extensive set of baselines.

Reviewers note the submission tackles a novel (DqhU) and important (UCgS) problem and introduces a novel approach (dRt2) which is evaluated on a comprehensive set of baselines, showing a convincing empirical performance (DqhU, UCgS).

This is a borderline case: Reviewers dRt2 and UCgs think it is marginally above the acceptance threshold, and Reviewer DqHu thinks it is marginally below the acceptance threshold. The main concerns center around three points:

1. Reviewers UCgS and DqhU both note the narrow scope of the evaluation datasets. The authors respond that OpinionQA and GlobalOpinionQA are to their knowledge the most comprehensive and relevant opinion datasets currently available, and that there are no long-form QA datasets suitable for benchmarking in a "diverse demographic groups" setting due to the prohibitive cost of building such a dataset. The issue remains the main issue in the way of a stronger acceptance recommendation for Reviewer UCgS (which was also echoed by Reviewer dRt2 in the reviewer-AC discussion), and Reviewer DqhU is worried that the current evaluation is not informative of GPO's generalizability to longer response generation (see their reproduced comment in Justification For Why Not Higher Score). There is merit to both sides of the argument: the lack of suitable long-form QA datasets to evaluate on is conceivably outside the authors' control, but it also leaves an important question unanswered, which reduces the potential practical impact of the proposed approach. What tilts the balance in favor of the submission here is that there is consensus among reviewers that the problem being tackled is under-explored and important, and that makes the paper valuable despite the limitations of its evaluation.

2. Reviewer dRt2 mentions in the reviewer-AC discussion that the main weakness preventing a higher score is that the proposed approach does not feel particularly novel to them. The authors stress out that GPO is not an in-context finetuning approach, in that it does not adapt the pretrained LLM's parameters. That being said, it does make use of the in-context training principle, albeit with a separate Transformer trained on top of the pretrained LLM's embeddings. Once again, given the under-explored nature of the problem and its importance, the submission remains valuable, as evidenced by Reviewer dRt2's weak acceptance recommendation.

3. Reviewer DqhU finds the exposition of the proposed approach confusing (see their reproduced comment in Justification For Why Not Higher Score). The authors did make a significant effort at clarifying all reviewers' questions, but this is insufficient for Reviewer DqhU. I agree with the reviewer to an extent: reading the submission I also found it hard to parse how the GPO Transformer interfaces with the pretrained LLM, and the submission could benefit from a revision focusing on making the implementation of GPO as clear and accessible as possible.

**Justification For Why Not Higher Score:**

The submission still has room for improvement in terms of clarity, and I agree with Reviewer DqHu's assessment that many details on how the GPO Transformer interfaces with the pretrained model are hard to decipher, even in the updated version of the manuscript.

For transparency I am reproducing Reviewer DqHu's final comment which I believe was mistakenly not made visible to the authors:

> I have spent an hour to carefully read the author response, other reviewers' comments and the revised paper method section again. However, I am still very confused about the proposed method:

> The author responded that the method proposes to train another transformer for the group preference prediction using the supervised data. Then how should we use the prediction of this new transformer model to steer the original instruction-tuned model like llama2-chat to generate responses considering the preference of the corresponding group? This part of the connection between the new transformer model and the original base model is missing in the method section. Besides, what is the architecture of this new transformer model? How many layers is it and what is hidden dimension? Is it initialized from a pretrained model? If so, what is it? Such important information is still missing.

> The dataset or task used in this paper is very rare and new. I am wondering whether the output of this dataset is just one choice among several options. Is the output short text? If so, how could we assure that this method can really generalize to steer long response generation to group preference?

> In the author response, they mention that the proposed method does not alter the base model weights so it does not affect the capabilities on other tasks. However, in this case, how should we determine when to use the group preference model to steer the LLM output? Do we have a binary decision maker model to determine when to steer the LLM or just use the original output from the base LLM?

> TBH, I am not familiar with the datasets and task tackled by this paper and I still cannot get how important it is, especially to the real application of LLMs in industry. If it is important to research community, I am ok with publishing it.

**Justification For Why Not Lower Score:**

One of the submission's strengths is that it tackles a new and important contemporary problem; this strength outweighs its flaws for a majority of the reviewers. Reviewer DqHu, who thinks the submission is marginally below the acceptance threshold, also remains open to accepting on this basis (see their reproduced comment in the Justification For Why Not Higher Score).

---

### Decision · Program_Chairs · 2024-01-16

Accept (poster)